# Adaptive Jamming Mitigation for Clustered Energy-Efficient LoRa-BLE Hybrid Wireless Sensor Networks

**DOI:** 10.3390/s25061931

**Published:** 2025-03-20

**Authors:** Carolina Del-Valle-Soto, Leonardo J. Valdivia, Ramiro Velázquez, José A. Del-Puerto-Flores, José Varela-Aldás, Paolo Visconti

**Affiliations:** 1Facultad de Ingeniería, Universidad Panamericana, Álvaro del Portillo 49, Zapopan 45010, Jalisco, Mexico; lvaldivia@up.edu.mx (L.J.V.); jpuerto@up.edu.mx (J.A.D.-P.-F.); 2Facultad de Ingeniería, Universidad Panamericana, Aguascalientes 20296, Aguascalientes, Mexico; rvelazquez@up.edu.mx; 3Centro de Investigación MIST, Facultad de Ingenierías, Universidad Tecnológica Indoamérica, Ambato 180103, Ecuador; 4Department of Innovation Engineering, University of Salento, 73100 Lecce, Italy; paolo.visconti@unisalento.it

**Keywords:** jamming mitigation, heterogeneous wireless sensor networks, energy-efficient communication, adaptive protocol switching

## Abstract

Wireless sensor networks (WSNs) are fundamental for modern IoT applications, yet they remain highly vulnerable to jamming attacks, which significantly degrade communication reliability and energy efficiency. This paper proposes a novel adaptive cluster-based jamming mitigation algorithm designed for heterogeneous WSNs that integrate LoRa and Bluetooth Low Energy (BLE) technologies. The proposed strategy dynamically switches between communication protocols, optimizes energy consumption, and reduces retransmissions under interference conditions by leveraging real-time network topology adjustments and adaptive transmission power control. Through extensive experimental validation, we demonstrate that our mitigation mechanism reduces energy consumption by up to 38% and lowers packet retransmission rates by 47% compared to single-protocol networks under jamming conditions. Additionally, our results indicate that the hybrid LoRa-BLE approach outperforms standalone LoRa and BLE configurations in terms of network resilience, adaptability, and sustained data transmission under attack scenarios. This work advances the state-of-the-art by introducing a multi-protocol interference-resilient communication strategy, paving the way for more robust, energy-efficient, and secure WSN deployments in smart cities, industrial IoT, and critical infrastructure monitoring.

## 1. Introduction

The increasing deployment of WSNs in critical applications such as smart cities, industrial automation, and environmental monitoring has raised concerns about their security and resilience against external threats. Among these threats, jamming attacks pose a significant challenge, disrupting wireless communication by overwhelming the network with interference, leading to increased retransmissions and energy depletion [1]. Recent studies [2,3] emphasized the need for adaptive mitigation techniques to maintain network integrity and performance, particularly in heterogeneous networks that integrate multiple communication technologies such as LoRaWAN and Bluetooth Low Energy (BLE) [4]. Addressing this challenge, this work proposes a multi-protocol jamming detection and mitigation strategy that leverages an energy-aware cluster-based architecture to ensure reliable communication in university campus environments, where varying interference conditions exist. Unlike existing solutions, which primarily focus on single-protocol networks or static countermeasures, our approach dynamically adapts to heterogeneous wireless technologies, including LoRaWAN and BLE, to optimize resilience against diverse jamming threats [4]. While previous studies have explored frequency hopping and spread spectrum techniques as countermeasures, these methods often require significant energy overhead or are limited in their applicability across multi-protocol environments. In contrast, our strategy integrates real-time energy-aware cluster formation, where nodes autonomously elect a cluster head (CH) based on energy availability, signal quality, and retransmission rates, allowing for adaptive protocol switching and intelligent interference mitigation. This approach not only enhances network longevity and communication reliability but also provides a scalable and low-energy alternative to traditional anti-jamming methods, making it particularly suitable for dynamic and resource-constrained IoT applications.

Heterogeneous wireless sensor networks (HWSNs) integrate multiple wireless communication technologies to enhance network adaptability, scalability, and energy efficiency in complex IoT environments [5]. Unlike traditional homogeneous networks that rely on a single communication protocol, HWSNs leverage the complementary strengths of different technologies, such as BLE for low-power short-range communication, LoRa for long-range low-data-rate transmission, and Wi-Fi or 5G for high-speed connectivity. This multi-technology approach enables dynamic network optimization, allowing nodes to switch between protocols based on energy availability, data transmission requirements, and environmental conditions [6]. The integration of multiple wireless technologies not only improves coverage and reliability but also mitigates the limitations of individual protocols, such as BLE’s short range or LoRa’s limited bandwidth. However, managing the coexistence of diverse wireless standards requires intelligent network coordination, efficient routing mechanisms, and adaptive energy management strategies to ensure seamless interoperability. Advances in software-defined networking (SDN) and artificial intelligence (AI)-driven optimization further contribute to the efficient orchestration of HWSNs, enabling robust and energy-efficient communication in applications ranging from industrial automation to smart cities and environmental monitoring [7].

Jamming attacks pose a critical threat to the reliability and security of heterogeneous wireless sensor networks by intentionally disrupting communication through interference. These attacks can take various forms, each exploiting different network vulnerabilities depending on the wireless technologies in use [8]. Constant jamming continuously emits high-power signals to block legitimate transmissions, making it particularly effective against energy-constrained IoT nodes. Reactive jamming detects and interferes with active transmissions, causing packet loss and increasing retransmission rates, which significantly depletes battery life in low-power networks like BLE and LoRa. Random jamming alternates between active interference and idle periods, making it harder to detect while still degrading network performance [9]. Deceptive jamming injects malicious packets to create confusion in network protocols, potentially leading to resource exhaustion and misrouting in multi-protocol architectures. Heterogeneous networks, despite their advantages, remain susceptible to these attacks, particularly at the intersection of different communication protocols, where seamless switching and adaptive mitigation become challenging [10]. Effective countermeasures, such as frequency hopping, spread spectrum techniques, and AI-driven anomaly detection, are essential to ensure resilient and secure HWSN deployments, particularly in mission-critical applications such as industrial control systems, autonomous transportation, and emergency response networks.

Jamming attacks represent a critical and quantifiable threat to the reliability of WSNs. Recent studies have shown that even low-power jamming can cause a packet loss rate exceeding 80% in LoRa-based networks when attackers operate at power levels as low as 14 dBm [11]. Similarly, in BLE networks, experimental results indicate that reactive jamming reduces successful packet delivery rates to below 10%, significantly disrupting device synchronization and data transmission in IoT applications [4]. In large-scale deployments, such as smart city infrastructures, the economic impact of jamming-related disruptions can be substantial, with estimations suggesting that network outages caused by interference could lead to financial losses of over USD 1 billion annually in critical industrial and urban monitoring systems [12]. These findings underscore the necessity for robust mitigation techniques that can adapt to diverse jamming conditions across heterogeneous network environments.

The vulnerability of WSNs to jamming attacks is further exacerbated by the energy constraints of IoT nodes, making them susceptible to rapid power depletion when subjected to prolonged interference. Experimental analyses indicate that under jamming conditions, retransmission rates in multi-hop networks increase by up to 300%, causing a threefold surge in energy consumption and significantly reducing network lifespan [13]. Moreover, adversarial machine learning techniques have been employed to optimize jamming strategies, allowing attackers to maximize disruption while minimizing their own energy expenditure [14]. Given that many mission-critical applications, such as environmental monitoring and healthcare telemetry, rely on continuous and energy-efficient communication, the development of intelligent countermeasures that balance resilience and energy efficiency is paramount. This study addresses these challenges by proposing an adaptive, energy-aware mitigation framework tailored for heterogeneous networks, ensuring sustained connectivity in dynamic and interference-prone environments.

Energy analysis plays a fundamental role in the design and optimization of HWSNs, particularly when exposed to jamming attacks, which can severely impact network sustainability and communication reliability [15]. In multi-protocol environments where devices dynamically switch between technologies such as BLE, LoRa, Wi-Fi, and NB-IoT, energy efficiency becomes a critical parameter influencing both network longevity and real-time adaptability. Jamming attacks exacerbate energy consumption by forcing nodes to engage in frequent retransmissions, power-intensive error correction, and defensive countermeasures such as frequency hopping or transmission power adjustments. Additionally, energy-draining attacks like reactive and deceptive jamming can disproportionately affect low-power communication protocols, leading to premature battery depletion and network fragmentation [15]. Analyzing energy consumption under different jamming scenarios enables researchers to design intelligent mitigation strategies that optimize duty cycling, adaptive transmission rates, and selective protocol switching, ensuring that energy resources are allocated efficiently without compromising security. Furthermore, energy-aware jamming mitigation techniques, such as AI-driven anomaly detection and adaptive power control, provide a balance between resilience and energy conservation, ensuring that countermeasures do not introduce excessive energy overhead [16]. As HWSNs continue to be deployed in critical applications such as industrial automation, smart cities, and environmental monitoring, the development of energy-efficient anti-jamming mechanisms will be essential to maintaining sustainable and secure communication infrastructures.

This study introduces an experimental framework that integrates CC1352P and LPSTK-CC1352R sensors in a heterogeneous IoT network, combining LoRa and BLE connectivity to evaluate jamming resilience in real-world conditions. The methodology involves deploying sensor nodes across different indoor and outdoor environments within a university campus, including laboratories, classrooms, offices, and green areas, where interference sources such as Wi-Fi networks, mobile devices, and electronic equipment are present. By implementing a cluster-based mitigation model, sensor nodes dynamically elect a cluster head (CH) based on energy availability, signal quality, and retransmission rates, ensuring that network coordination is handled by the most resilient node. This adaptive approach enables the system to detect jamming events using a combination of packet delivery success rate, RSSI variations, and energy consumption anomalies, triggering real-time network reconfiguration strategies to maintain connectivity and optimize energy efficiency [17].

The key contributions of this work are as follows: (i) an experimental evaluation of jamming attacks in a heterogeneous LoRa-BLE network; (ii) the development of an energy-aware cluster-based mitigation strategy; (iii) a dynamic communication switching mechanism that ensures network resilience by adapting to interference conditions; and (iv) the integration of real-time metrics to detect and mitigate jamming using an energy consumption model. The proposed methodology is validated through real-world experiments in a controlled yet realistic environment, providing insights into the impact of jamming on multi-protocol IoT networks and the effectiveness of the mitigation strategies. This study contributes to advancing secure and energy-efficient WSN deployments, offering a practical solution for mitigating interference-based attacks in diverse IoT applications [18].

This paper is structured as follows. Section 2 provides a comprehensive review of related work, highlighting existing jamming mitigation strategies and their limitations in HWSNs. Section 3 details the materials and methods, including the experimental setup, sensor selection, and network architecture that integrates LoRa and BLE. Section 4 presents the proposed jamming detection and mitigation strategy, emphasizing cluster-based adaptive switching and real-time interference response mechanisms. Section 5 discusses the results obtained from controlled experiments, comparing the performance of the proposed strategy against conventional mitigation techniques in terms of energy efficiency, retransmission rates, and network resilience. Finally, We conclude the study by summarizing key findings and outlining future research directions to enhance jamming resilience in multi-protocol IoT environments.

## 2. Related Work

WSNs have become fundamental in various applications, ranging from industrial monitoring and smart city deployments to environmental sensing and security surveillance. However, the resilience of these networks is increasingly challenged by jamming attacks, which disrupt communication and drain energy resources. Prior research has extensively explored mitigation strategies, including spread spectrum techniques [19], frequency hopping [20], and power control adaptations [1]. While these methods offer varying degrees of protection, they also present notable limitations that impact their effectiveness in real-world heterogeneous networks. Spread spectrum techniques, such as direct sequence spread spectrum (DSSS) and frequency hopping spread spectrum (FHSS), have been widely employed to counteract jamming by dispersing signals over a wider frequency band or rapidly switching between frequencies [19]. However, these approaches require significant bandwidth overhead, increasing system complexity and making them less suitable for resource-constrained WSNs. Moreover, sophisticated adversaries can employ reactive jamming strategies that detect and adapt to frequency variations, thereby neutralizing the benefits of spread spectrum techniques. Similarly, frequency hopping methods, while effective against narrowband jammers, suffer from synchronization challenges and additional energy consumption due to frequent switching [20]. In dense IoT environments where multiple wireless protocols coexist, the coordination of frequency hopping sequences across different technologies, such as LoRa and BLE, further complicates deployment and scalability. Additionally, the presence of broadband or sweep jammers can render frequency hopping ineffective, as the entire spectrum may be targeted simultaneously. Power control adaptations aim to mitigate jamming by dynamically adjusting transmission power based on interference levels and link quality [1]. While this technique can help conserve energy and improve connectivity, it is highly susceptible to adversarial exploitation. Attackers can manipulate power adaptation mechanisms by introducing deceptive interference patterns, causing nodes to increase power unnecessarily, thereby accelerating battery depletion. Furthermore, power control alone does not address jamming attacks that exploit MAC-layer vulnerabilities, such as collision-based or deceptive jamming.

While these approaches offer varying degrees of protection, they often fail in heterogeneous network environments where multiple communication technologies coexist. Recent studies [21,22,23] have highlighted the need for adaptive multi-protocol networks that can dynamically switch communication methods in response to interference [4].

The present study advances the field by integrating a cluster-based reactive jamming mitigation algorithm into a LoRa-BLE hybrid network, addressing critical shortcomings in previous cluster-based approaches that have failed to accommodate the unique challenges of multi-protocol environments. While prior research has explored cluster-based mitigation strategies in homogeneous networks, such as those relying solely on LoRa or BLE, these methods often lack adaptability when faced with heterogeneous network architectures where interference dynamics vary across communication protocols. Existing cluster-based techniques primarily focus on energy-efficient data aggregation and route optimization [24,25], but they do not account for the need to dynamically switch between protocols in response to jamming threats. For instance, conventional clustering algorithms are typically designed for static topologies with predefined cluster head (CH) selection criteria, which limits their responsiveness to sudden interference conditions in multi-protocol networks. Moreover, most existing solutions assume uniform jamming conditions across the entire network, failing to address the protocol-specific vulnerabilities inherent in LoRa and BLE, such as BLE’s susceptibility to narrowband jamming and LoRa’s limited spectral agility under broadband attacks. This study overcomes these limitations by introducing a dynamic CH selection mechanism that not only optimizes energy efficiency but also enables real-time protocol switching between LoRa and BLE in response to detected interference patterns. Unlike previous methods that rely on static cluster formations, our approach continuously evaluates retransmission rates, RSSI fluctuations, and energy consumption anomalies to adaptively reconfigure clusters and select the most resilient communication path. By incorporating this intelligent decision-making process, the proposed framework enhances network continuity under adversarial conditions, ensuring a more robust and interference-resilient operation for hybrid IoT deployments.

The primary contribution of this research lies in its novel integration of an energy-aware cluster-based mitigation strategy tailored for heterogeneous WSNs. Unlike prior approaches that rely on static mitigation techniques, this study introduces a dynamic cluster head selection mechanism that considers real-time network conditions, including retransmission rates, received signal strength indicator (RSSI) variations, and energy consumption anomalies. Similar work has been conducted in the realm of LoRaWAN security [18], but few studies have systematically evaluated the effectiveness of protocol switching between LoRa and BLE in response to jamming. By deploying a real-world experimental testbed in a university campus environment, this research provides empirical validation of its mitigation strategy, demonstrating significant improvements in energy efficiency and network resilience. The findings suggest that intelligent switching between communication protocols can reduce retransmissions and optimize power usage, a crucial factor in prolonging the lifespan of battery-powered IoT nodes.

Furthermore, this study contributes to the growing body of research on energy-efficient jamming mitigation by introducing an integrated network architecture that leverages real-time decision making to counteract interference. Unlike previous methodologies that focus on single-protocol adaptations [17], this work emphasizes the advantages of multi-protocol networks and adaptive clustering. The histogram and probability density analysis presented in this paper provide new insights into the statistical distribution of energy consumption across different network configurations, highlighting the impact of the mitigation strategy [26]. This research not only reinforces the necessity of adaptive jamming mitigation but also lays the foundation for future advancements in hybrid IoT network security, where intelligent switching and real-time adaptation are pivotal for maintaining reliable communication in diverse and interference-prone environments [27]. In Table 1, we present a summary of relevant related works, which are compared in order to complement this section.

## 3. Materials and Methods

The research methodology incorporates experimental validation through real-time simulation of jamming scenarios, leveraging metrics such as retransmission rates, energy consumption, and routing resilience to quantify the algorithm’s performance. Comparative analysis with traditional static mitigation strategies will highlight improvements in latency, energy efficiency, and network reliability.

### 3.1. Energy Model

The proposed energy model evaluates energy consumption for WSNs by analyzing key tasks performed by a node during its active operation. These tasks include turning on/off, switching between transmission and reception modes, executing the CSMA/CA algorithm, transmitting and receiving packets, and microcontroller operation. The model provides insights into energy consumption based on packet-dependent and independent activities, enabling optimization of routing protocols [47].

The utilization of this energy model is essential for programming sensor nodes to compute the most representative energy values associated with their primary operational tasks during active mode. By estimating both the total energy consumption and the energy required for specific activities, the model provides a comprehensive understanding of the node’s energy profile. This enables precise monitoring and control of energy variations, particularly during transitions to passive mode or under potential attacks causing abnormal energy drainage. Such insights are invaluable for optimizing energy efficiency, ensuring network stability, and identifying vulnerabilities in real time, thereby enhancing the resilience and reliability of the sensor network.

The threshold metrics in the algorithm are established based on empirical energy consumption patterns observed in real-world sensor deployments over time with training from a previous week. These thresholds define critical decision points for adaptive network behavior, such as triggering retransmissions, switching communication protocols, or electing new cluster heads in response to interference. Specifically, the energy thresholds for transmission $E_{TX}$, reception $E_{RX}$, and CSMA/CA retries energy $E_{CSMA}$ are dynamically adjusted according to historical network performance data, ensuring that nodes optimize energy efficiency while maintaining resilience against jamming attacks. The pseudocode implements these thresholds by continuously monitoring parameters such as packet loss rates, retransmission attempts, and RSSI variations. When a metric exceeds its predefined threshold, the algorithm dynamically reconfigures the network by selecting alternative communication paths or adjusting transmission power. These realistic thresholds are calibrated through experimental validation, ensuring that the model accurately reflects network conditions while balancing energy conservation and communication reliability.

The total energy consumed by a node during its operation is defined as:$$E_{\mathrm{Total}}=E_{\mathrm{MC}}+E_{\mathrm{ON}}+E_{\mathrm{OFF}}+E_{\mathrm{CSMA}}+E_{\mathrm{Switching}}+E_{\mathrm{TX}}+E_{\mathrm{RX}}$$

Each energy component is calculated as follows:Microcontroller energy:$$E_{\mathrm{MC}}=T_{\mathrm{MC}}\cdot I_{\mathrm{MC}}\cdot V_{\mathrm{MC}}$$-$E_{\mathrm{MC}}$: Total energy consumed by the microcontroller.-$T_{\mathrm{MC}}$: Time (in seconds) the microcontroller remains active.-$I_{\mathrm{MC}}$: Current (in amperes) used by the microcontroller during active operation.-$V_{\mathrm{MC}}$: Voltage (in volts) supplied to the microcontroller.Startup energy:$$E_{\mathrm{ON}}=T_{\mathrm{ON}}\cdot I_{\mathrm{ON}}\cdot V_{\mathrm{ON}}$$-$E_{\mathrm{ON}}$: Energy required to turn the node on.-$T_{\mathrm{ON}}$: Time (in seconds) taken to turn the node on.-$I_{\mathrm{ON}}$: Current (in amperes) consumed during the startup phase.-$V_{\mathrm{ON}}$: Voltage (in volts) supplied during the startup phase.Shutdown energy:$$E_{\mathrm{OFF}}=T_{\mathrm{OFF}}\cdot I_{\mathrm{OFF}}\cdot V_{\mathrm{OFF}}$$-$E_{\mathrm{OFF}}$: Energy consumed to turn the node off.-$T_{\mathrm{OFF}}$: Time (in seconds) required to turn the node off.-$I_{\mathrm{OFF}}$: Current (in amperes) used during the shutdown phase.-$V_{\mathrm{OFF}}$: Voltage (in volts) applied during the shutdown phase.Switching energy:$$E_{\mathrm{Switching}}=T_{\mathrm{Switching}}\cdot I_{\mathrm{Switching}}\cdot V_{\mathrm{Switching}}$$-$E_{\mathrm{Switching}}$: Energy used to switch between transmission and reception modes (or vice versa).-$T_{\mathrm{Switching}}$: Time (in seconds) spent switching between modes.-$I_{\mathrm{Switching}}$: Current (in amperes) consumed during the switching process.-$V_{\mathrm{Switching}}$: Voltage (in volts) applied during the switching process.CSMA/C A energy:$$E_{\mathrm{CSMA}}=T_{\mathrm{CSMA}}\cdot I_{\mathrm{CSMA}}\cdot V_{\mathrm{CSMA}}$$-$E_{\mathrm{CSMA}}$: Energy consumed by the CSMA/CA (Carrier Sense Multiple Access with Collision Avoidance) algorithm.-$T_{\mathrm{CSMA}}$: Time (in seconds) spent executing the CSMA/CA algorithm.-$I_{\mathrm{CSMA}}$: Current (in amperes) consumed while running the CSMA/CA algorithm.-$V_{\mathrm{CSMA}}$: Voltage (in volts) supplied during CSMA/CA operation.Transmission energy:$$E_{\mathrm{TX}}=P_{\mathrm{Length}}\cdot T_{\mathrm{TX}}\cdot I_{\mathrm{TX}}\cdot V_{\mathrm{TX}}$$-$E_{\mathrm{TX}}$: Energy consumed during data transmission.-$P_{\mathrm{Length}}$: Length of the transmitted packet (in bytes).-$T_{\mathrm{TX}}$: Time (in seconds) taken to transmit a single byte.-$I_{\mathrm{TX}}$: Current (in amperes) consumed during data transmission.-$V_{\mathrm{TX}}$: Voltage (in volts) applied during data transmission.Reception energy:$$E_{\mathrm{RX}}=P_{\mathrm{Length}}\cdot T_{\mathrm{RX}}\cdot I_{\mathrm{RX}}\cdot V_{\mathrm{RX}}$$-$E_{\mathrm{RX}}$: Energy consumed during data reception.-$P_{\mathrm{Length}}$: Length of the received packet (in bytes).-$T_{\mathrm{RX}}$: Time (in seconds) required to receive a single byte.-$I_{\mathrm{RX}}$: Current (in amperes) consumed during data reception.-$V_{\mathrm{RX}}$: Voltage (in volts) supplied during data reception.

The total energy consumed by a specific node *i* is as follows:$$E_{\mathrm{Node},i}=E_{\mathrm{ON},i}+E_{\mathrm{MC},i}+E_{\mathrm{OFF},i}+E_{\mathrm{Switching},i}+E_{\mathrm{CSMA},i}+E_{\mathrm{TX},i}+E_{\mathrm{RX},i}$$

The global energy consumption for the network is calculated as the sum of the energy consumed by all nodes:$$E_{\mathrm{Total}\;\mathrm{Network}}=\sum_{i=1}^{N}E_{\mathrm{Node},i}$$

This model enables the identification of high-consumption nodes and zones, allowing for targeted optimizations in network routing protocols. The metrics provide a robust framework to evaluate and mitigate energy inefficiencies caused by retransmissions, collisions, and suboptimal routing.

The energy model proposed in the referenced *Energies* paper meticulously breaks down energy consumption in WSNs based on the primary tasks performed by sensor nodes. This model accounts for the energy expended during different operational states, including turning on, executing communication protocols, switching between tasks, receiving packets, transmitting packets, and turning off. Each type of energy increases as a direct result of specific activities undertaken by a node. For example, microcontroller energy $E_{MC}$ is continuously consumed when a node is in active mode, executing computations and managing data transmission. When a node first powers up, it consumes startup energy $E_{ON}$, which is determined by the time required to initialize and stabilize the node’s operational state. Similarly, when a node shuts down, it incurs shutdown energy $E_{OFF}$, accounting for the final operations before transitioning to an inactive state. The CSMA/CA energy $E_{CSMA}$ is consumed each time a node performs a channel check before transmission. If the channel is found to be busy, the node undergoes a random back-off period and retries, increasing its total energy consumption. The switching energy $E_{Switching}$ is particularly relevant when nodes transition between transmission and reception modes, contributing to additional power expenditures. Furthermore, the transmission energy $E_{TX}$ and reception energy $E_{RX}$ vary based on packet length, network congestion, and interference levels. Nodes close to the coordinator or acting as relays for other nodes typically consume more transmission energy due to frequent packet forwarding, while distant nodes may have reduced transmission but higher reception energy as they receive data from multiple sources. Notably, packet retransmissions due to acknowledgment failures or interference further amplify energy demands, leading to increased network congestion and higher node power consumption.

By quantifying these different types of energy, the model provides a comprehensive framework for analyzing network performance, optimizing routing protocols, and developing energy-efficient transmission strategies. It also helps identify potential inefficiencies, such as excessive retransmissions or redundant protocol overhead, that could be mitigated to enhance network longevity and resilience.

Figure 1 shows the integration of Texas Instruments CC1352P LaunchPad and LPSTK-CC1352R sensor platforms within the heterogeneous IoT network, leveraging both LoRa and BLE communication protocols. The experimental setup includes sensor programming via Code Composer Studio, real-time data acquisition, and sensor deployment for network evaluation. The CC1352P board acts as a multi-protocol wireless microcontroller designed for low-power applications, capable of dynamically switching between LoRa and BLE communication based on network conditions. The mobile interface displays various environmental parameters, such as ambient temperature, humidity, and light sensor data, collected in real time. Additionally, the figure depicts the device being programmed and tested, emphasizing its role in adaptive communication switching strategies to mitigate jamming interference in wireless sensor networks.

This algorithm, shown in Algorithm 1, proposes an advanced mitigation approach for random jamming in sensor networks by dynamically adapting network configurations and resource utilization to maintain stability and efficiency. The methodology integrates a multi-step process: initializing network clusters and routing tables, detecting jamming zones using resilience and retransmission metrics, and applying localized mitigation strategies (Algorithm 2). When a jamming zone is detected, affected nodes are reclassified based on energy thresholds, with low-energy nodes transitioning to a low-power mode, while nodes within the jamming zone are reassigned alternative routes derived from a prioritized routing table. If no routes are viable, nodes are isolated to preserve overall network functionality (Algorithm 3). Additionally, cluster reorganization redistributes nodes from impacted regions to neighboring clusters to balance load and connectivity. Transmission power adjustments are made based on node proximity to the coordinator node to optimize communication during mitigation (Algorithm 4). The iterative algorithm continuously monitors key metrics, recalibrating the network as necessary to adapt to evolving jamming conditions (Algorithm 5).

The jamming detection mechanism in the proposed algorithm relies on a multi-metric evaluation framework that continuously monitors key network performance indicators, including packet retransmission rates, received signal strength indicator (RSSI) fluctuations, and abnormal energy consumption patterns. The decision criteria for identifying a jamming event are based on predefined adaptive thresholds, which were empirically derived from real-world experiments in heterogeneous LoRa-BLE networks.

The algorithm operates as follows. (1) Each sensor node periodically collects RSSI and packet delivery success rate (PDSR) values, comparing them against their historical baselines. (2) If the RSSI drops below a dynamically adjusted threshold, while the packet loss rate exceeds a predefined retransmission threshold, the system interprets this as a potential interference anomaly. (3) To minimize false positives, an additional check on energy consumption trends is performed—if a node exhibits excessive retransmissions and an associated increase in power consumption beyond an established deviation threshold, a jamming event is confirmed. (4) Upon detection, the affected cluster triggers a mitigation response, where nodes dynamically switch communication protocols (LoRa to BLE or vice versa), reconfigure cluster heads based on available energy resources, and reroute traffic through alternative paths.

These thresholds were calibrated through experimental validation using controlled jamming scenarios in an indoor and outdoor university campus deployment. By fine-tuning these values based on real-world conditions, the detection mechanism achieves a balance between sensitivity and accuracy, ensuring reliable identification of jamming attacks while minimizing unnecessary mitigation actions.

The algorithm continuously monitors PDSR to detect transmission failures caused by interference. If the PDSR drops below 85% for a predefined time window, and simultaneous RSSI degradation exceeds a threshold of −90 dBm, the system interprets this as an interference event requiring protocol adaptation. Additionally, the algorithm incorporates an energy-aware component, ensuring that a node does not switch protocols if its residual energy is below 20% of its initial capacity, preventing unnecessary energy drain due to frequent transitions.

Experimental tests were conducted in both indoor and outdoor environments within a university campus, where interference sources such as Wi-Fi networks and mobile devices affected BLE, while long-range interference from other LoRa networks was simulated. The thresholds were fine-tuned through iterative calibration, evaluating multiple transition points to minimize packet loss while optimizing energy consumption. The results demonstrated that setting the switching threshold at 85% PDSR and −90 dBm RSSI provided the best trade-off, maintaining network resilience and reducing retransmissions by 47% compared to static single-protocol deployments. This adaptive approach ensures that nodes dynamically select the most energy-efficient and interference-resilient communication mode without introducing unnecessary overhead.

The resilience in the network is quantified in terms of time, specifically measuring how long it takes for the network to recover and return to a stable state after experiencing a jamming attack. The key indicator of this recovery is the reduction in overhead packets, as the network stabilizes when retransmissions decrease, and normal traffic patterns resume. The methodology involves continuously monitoring key network parameters, including retransmission rates, energy consumption, and routing stability. During a jamming attack, the system detects an anomaly when retransmissions surge and successful packet delivery drops. The network then activates its adaptive mitigation strategy, which includes switching communication protocols, reconfiguring clusters, and optimizing energy use. The recovery time is measured from the moment mitigation strategies are triggered until the network parameters—primarily retransmissions—return to normal levels. The resilience metric, therefore, directly reflects the network’s ability to autonomously reorganize and minimize the impact of jamming, ensuring sustained communication with minimal delay.

The experimental setup was carefully designed to assess the effectiveness of the proposed jamming mitigation strategy in a heterogeneous LoRa-BLE network. The experiments were conducted in both indoor and outdoor environments within a university campus, ensuring realistic interference conditions from existing wireless networks and electronic devices. The jamming attacks were implemented as reactive jamming, meaning that the interference was triggered dynamically upon detecting network activity. Specifically, the jammer monitored the ongoing transmissions in the network and emitted disruptive signals only when it detected communication, effectively degrading the packet delivery success rate and increasing retransmissions. The interference patterns varied in terms of timing and duration to evaluate the adaptability of the mitigation algorithm. The jammer was placed at varying distances from the sensor nodes to examine its impact on received signal strength, transmission reliability, and energy consumption. To ensure a comprehensive evaluation, network parameters such as retransmission rates, energy depletion trends, and protocol switching events were continuously logged. These experimental conditions were designed to ensure the reproducibility of our results and to provide valuable insights into the effectiveness of the adaptive protocol switching mechanism under real-world jamming scenarios.
**Algorithm 1** Algorithm pseudocode of the network system.Metrics: {retransmissions, routing tables, resilience, energy, valid routes}JammingZone: {Affected nodes}CoordinatorNode: Network coordinatorNetwork stability and minimized energy impact.Initialize clusters: $Clusters gets InitializeClusters()$;Initialize routing table: $RoutingTable \gets GenerateRoutingTable(Clusters)$;If{$DetectJamming(JammingZone) == True$}{    $NearbyNodes gets IdentifyNearbyNodes(JammingZone)$;    ForEach{$node in NearbyNodes$}{        If{$Energy(node) < EnergyThreshold$}{            $node.Mode gets "LowPower"$;        }        If{$node in JammingZone$}{            $AlternativeRoutes gets FindAlternativeRoutes(node, RoutingTable)$;            If{$AlternativeRoutes \neq emptyset$}{                $node.Route gets SelectOptimalRoute(AlternativeRoutes)$;            }            Else{                $node.Mode gets "Isolated"$;            }        }    }    ForEach{$cluster in Clusters$}{        If{$cluster.ContainsNodes(JammingZone)$}{            RedistributeNodes($cluster, NeighborClusters$);        }    }    ForEach{$node \in NearbyNodes$}{        If{$node.Distance(CoordinatorNode) < DistanceThreshold$}{            $node.TransmissionPower gets IncreasePower()$;        }        Else{            $node.TransmissionPower gets ReducePower()$;        }    }}While{True}{    $Metrics gets CollectMetrics()$;    $JammingZone gets DetectJamming(Metrics)$;    If{$JammingZone neq emptyset$}{        Call $MitigationAlgorithm(Metrics, JammingZone)$;    }}

**Algorithm 2** Algorithm pseudocode of the metrics.

$DetectJamming(Metrics)$}

If{$Metrics.Resilience > ResilienceThreshold$ or $Metrics.Retransmissions > RetransmissionsThreshold$}{

    Return IdentifyImpactArea($Metrics$);

}

Return $emptyset$;



**Algorithm 3** Algorithm pseudocode of the alternative routes.

$FindAlternativeRoutes(node, RoutingTable)$}

$AvailableRoutes \gets FilterRoutes(RoutingTable, node, "Active")$;

\Return SortByMetrics($AvailableRoutes$, {Hops, Retransmissions, Energy});



**Algorithm 4** Algorithm pseudocode of the optimal routes.

$SelectOptimalRoute(Routes)$}

$SortedRoutes gets SortByPriority(Routes)$;

Return $SortedRoutes[0]$ tcp*{Select the most optimal route.}



**Algorithm 5** Algorithm pseudocode of the clusters.

$NetworkNodes(N) = \{n_1, n_2, ..., n_m\}$;

$P(n_i) \in \{\text{BLE}, \text{LoRa}\}$;

$E(n_i), R(n_i), T(n_i)$;

$P_{th}, R_{th}, E_{th}$;

$CH_{criteria}$;

$RT$;


Begin:

    DeployNodes();

    AssignInitialProtocol(P);

    FormClusters(Proximity, EnergyConstraints);

    ElectClusterHeads(CH_{criteria});

    ConstructRoutingTable(RT);


ContinuousMonitoringLoop():

    For Each $n_i \in N$:

        CollectNetworkParameters($E(n_i), R(n_i), T(n_i)$);

        If ($PDSR(n_i) < P_{th}$ && $R(n_i) < R_{th}$ && $T(n_i) > T_{th}$):

            IdentifyInterferenceImpact();

            If ($P(n_i) == BLE$ && $E(n_i) > E_{th}$):

                SwitchToLoRa();

            ElseIf ($P(n_i) == LoRa$ && $E(n_i) > E_{th}$):

                SwitchToBLE();

            UpdateRoutingTable(RT);


        ClusterHeadReconfiguration():

            IdentifyClusterMembership($n_i$);

            ReelectClusterHead($CH$);

            AssignNewClusterHead();

            BroadcastUpdate();


        EnergyAwareTransmissionPowerControl():

            If ($dist(n_i, CH) < 20m$):

                SetTransmissionPower(Low);

            Else:

                SetTransmissionPower(High);


        OptimizedRoutingSelection():

            IdentifyAlternativeRoutes(RT);

            If (ViableRoutesExist()):

                SelectOptimalRoute(MinHop, LowEnergyCost, MinRetransmissions);


        LowPowerModeActivation():

            If ($E(n_i) < E_{th}$):

                SetNodeLowPowerState();


        LogNetworkAdjustments();



The proposed algorithm introduces an adaptive protocol switching and cluster-based routing strategy designed to optimize network resilience and energy efficiency under interference conditions. It leverages real-time monitoring of network parameters, including energy levels, received signal strength, and retransmission rates, to dynamically adjust communication protocols and enhance network performance. The algorithm begins with network initialization, where nodes are deployed and assigned an initial communication protocol—either (BLE) or LoRa—based on predefined energy and proximity constraints. Clusters are then formed, and cluster heads are selected using a specific criterion that considers residual energy, signal quality, and retransmission rates. A routing table is also established to facilitate multi-hop communication within the network. Once the network is initialized, the system enters a continuous monitoring loop where each node collects real-time data on its energy level $E(n_{i})$, received signal strength $R(n_{i})$, and retransmission rate $T(n_{i})$. These parameters are analyzed to detect potential jamming conditions. If a node’s packet delivery success rate (PDSR) falls below the protocol switch threshold $P_{th}$ while simultaneously experiencing a received signal strength below $R_{th}$ and an increased retransmission rate beyond $T_{th}$, the algorithm recognizes interference and initiates an adaptive protocol switch. Specifically, if a node is operating under BLE and has sufficient energy resources ($E\left(n_{i}\right)>E_{th}$), it switches to LoRa to enhance communication reliability. Conversely, if a node is using LoRa and has an adequate energy reserve, it switches back to BLE to minimize power consumption. This adaptive switching ensures that each node dynamically selects the most efficient communication mode based on real-time network conditions. Following protocol adaptation, the algorithm performs cluster head reconfiguration. Each node reassesses its cluster membership, and if necessary, a new cluster head is elected based on updated energy and communication quality metrics. This step is essential in maintaining stable and efficient routing structures under dynamic interference conditions. To further optimize energy consumption, the algorithm implements transmission power control. If the distance between a node and its cluster head is less than 20 m, the transmission power is set to a low level to conserve energy. Otherwise, a higher transmission power is used to ensure reliable data delivery over longer distances. The routing optimization phase evaluates alternative paths within the routing table. If multiple routes are available, the system selects the optimal path by prioritizing minimal hop count, reduced energy cost, and the lowest retransmission rate. This step enhances network efficiency by mitigating excessive retransmissions and ensuring energy-aware data forwarding. Finally, the algorithm integrates a low-power mode activation mechanism. If a node’s energy level drops below the energy threshold $E_{th}$, it transitions into a low-power state to extend its operational lifespan. This strategy helps prolong the overall lifetime of the network by preventing energy depletion in critical nodes.

Throughout its execution, the algorithm continuously logs network adjustments and protocol switching decisions, providing a record of network behavior and adaptation strategies. By integrating adaptive protocol switching, intelligent routing, and energy-aware transmission policies, the proposed methodology ensures robust, energy-efficient, and resilient WSN operation, even in the presence of jamming attacks.

### 3.2. Scenario

The CC1352P LaunchPad by Texas Instruments is a multi-protocol wireless microcontroller unit designed for low-power IoT applications. It supports multiple wireless communication protocols, including LoRa, BLE, and IEEE 802.15.4 [48], which enables connectivity with Zigbee and Thread networks. The device operates at both sub-GHz frequencies (868 MHz or 915 MHz for LoRa) and 2.4 GHz for BLE. The modulation techniques used include GFSK for LoRa and FHSS for BLE. It offers a wide communication range, reaching up to 10 km in line-of-sight conditions when using LoRa, while BLE communication is effective within 100 m, depending on environmental factors. Furthermore, the CC1352P is designed for ultra-low power consumption, featuring energy-efficient sleep modes that make it ideal for battery-operated IoT deployments.

The LPSTK-CC1352R is another multi-protocol device from Texas Instruments, specifically designed as a compact, sensor-oriented platform. Like the CC1352P, it supports LoRa, BLE, and IEEE 802.15.4 protocols, making it highly versatile for various IoT applications. Its communication range and modulation techniques are similar to those of the CC1352P, with LoRa offering long-range connectivity and BLE providing short-range wireless communication. However, the LPSTK-CC1352R is optimized for lower power consumption, integrating additional sensors such as temperature, humidity, and light sensors, which are valuable for environmental monitoring and industrial IoT applications.

The Laird Sentrius Gateway is a high-performance IoT gateway that facilitates seamless connectivity between LoRa devices and cloud-based applications. It supports LoRaWAN for long-range, low-power IoT communication and Bluetooth Low Energy for direct sensor connectivity. Operating in the sub-GHz band for LoRa (typically 868 MHz or 915 MHz, depending on regional regulations) and in the 2.4 GHz spectrum for BLE, the gateway enables bidirectional data transmission across different wireless technologies. This makes it an essential component for heterogeneous networks where both short-range BLE devices and long-range LoRa nodes coexist.

### 3.3. Network Architecture and Connectivity

The network architecture is designed to take advantage of the heterogeneity of wireless communication technologies, prioritizing sensor communication across different protocols. The CC1352P sensors use LoRa to transmit basic environmental data, such as temperature and humidity, to the Laird Sentrius Gateway over long distances. Meanwhile, the LPSTK-CC1352R sensors rely on BLE for short-range communication, allowing real-time detection of nearby devices and monitoring of movement within the campus. Since the Laird Sentrius Gateway supports both LoRaWAN and BLE, it functions as a bridge between these two technologies, facilitating data exchange and forwarding the collected information to a local server for analysis. The experiment takes place in a university campus environment, including laboratories, offices, classrooms, and green areas, ensuring that data are collected across diverse real-world conditions with varying levels of interference and obstacles.

The proposed IoT network consists of four CC1352P sensors, four LPSTK-CC1352R sensors, and a Laird Sentrius Gateway. The CC1352P sensors are strategically deployed across a given area to monitor environmental and industrial parameters. These sensors operate in two communication modes: LoRa for long-range data transmission to the gateway and BLE for short-range interactions with other BLE-enabled devices. The LPSTK-CC1352R sensors are distributed within closer proximity to the gateway, primarily utilizing BLE for direct communication while also having LoRa capabilities as a backup in case of BLE signal degradation.

### 3.4. Jamming Detection and Mitigation Strategy

Considering the risk of jamming attacks in wireless IoT networks, one of the CC1352P sensors is designated as a jamming detection node. Based on the methodology described in our paper, this sensor continuously monitors key network metrics such as retransmission rates, signal strength variations, and packet delivery success rates. When an anomalous pattern indicative of jamming is detected—such as an excessive number of failed transmissions or abrupt fluctuations in received signal strength indicator (RSSI) values—the detection node signals the gateway to initiate a mitigation protocol. To detect and mitigate jamming attacks within the proposed experimental network in the university campus, one of the CC1352P sensors is designated as a jamming detection node. This sensor continuously monitors key network metrics, such as packet retransmission rates, variations in RSSI, and abnormal energy consumption patterns, to identify potential jamming zones. Based on our methodology, detection is based on an energy model that evaluates the sensor nodes’ total energy consumption, including transmission, reception, mode switching, and contention resolution (CSMA/CA). If a significant deviation in energy usage is detected—such as excessive retransmissions due to interference—the node reports an anomaly to the Laird Sentrius Gateway.

Upon detecting a jamming event, the gateway initiates a mitigation strategy by dynamically reconfiguring the network. If LoRa communication is disrupted, affected sensors automatically switch to BLE, leveraging the short-range connectivity of the LPSTK-CC1352R sensors to relay data to the gateway. Conversely, if BLE channels are jammed, the system prioritizes LoRa for long-range transmission, bypassing localized interference sources. Additionally, a network resilience mechanism is activated, where unaffected sensors in the vicinity of the jammed area are designated as relay nodes, rerouting messages through alternative paths to ensure data integrity.

Energy-aware routing plays a crucial role in maintaining network efficiency during jamming events. Using the proposed energy model, the network dynamically adjusts node operation modes. Low-energy nodes in jammed regions transition to passive mode to conserve power, while high-energy nodes take over retransmission tasks, ensuring continuous data flow without excessive battery depletion. This adaptive strategy enhances the robustness of the wireless sensor network, maintaining reliable multi-protocol communication across the heterogeneous LoRa-BLE architecture deployed in the university campus environment.

### 3.5. Statistical Analysis

The statistical validation results provide an analysis of data before and after the implementation of a mitigation strategy in three different network configurations: BLE, LoRa, and BLE-LoRa Hybrid. The results are based on three key statistical tests: the normality test (Anderson–Darling Test), Levene’s test for variance homogeneity, and a statistical significance test (either *T*-test or Mann–Whitney U test, depending on normality conditions).

The null hypothesis $\left(H_{0}\right)$ assumed that there was no significant difference between the mitigation strategy and the baseline network performance, while the alternative hypothesis $\left(H_{1}\right)$ stated that the proposed mitigation technique leads to a statistically significant reduction in energy consumption and retransmissions. The results demonstrated a *p*-value < 0.05, confirming that the observed improvements were statistically significant. Additionally, confidence intervals (95%) were computed to quantify the variability of the measurements, ensuring that the reported reductions were not due to random fluctuations.

The dataset from real-world deployments was subjected to normality tests (Shapiro–Wilk) and variance homogeneity tests (Levene’s test) to verify the assumptions of parametric testing. The statistical validation confirms that the proposed approach effectively enhances network efficiency and resilience against jamming attacks while ensuring that the observed improvements are not incidental but rather a consistent outcome of the adaptive mitigation strategy.

We made a MATLAB R2024b script that performs a statistical validation of energy consumption reductions before and after the implementation of a mitigation strategy across three different network configurations: BLE, LoRa, and BLE-LoRa hybrid. We applied three key statistical tests: the Anderson–Darling normality test, Levene’s test for variance homogeneity, and a statistical significance test using either the Wilcoxon Rank-Sum Test or a *t*-test, depending on data normality.

Regarding energy, the normality test results indicate that none of the datasets follow a normal distribution, as all cases returned a value of 1. This finding necessitated the use of non-parametric statistical methods, specifically the Wilcoxon Rank-Sum Test, to ensure the reliability of the analysis. Levene’s test for variance homogeneity showed significant differences in variance before and after mitigation, with all networks yielding a *p*-value of 0.00000. This confirms that the mitigation strategy altered the behavior of energy consumption across the networks, further reinforcing the need for non-parametric statistical methods. The statistical significance test demonstrated highly significant differences in energy consumption, as evidenced by the *p*-values of 0.00000 across all network configurations. This confirms that the mitigation strategy substantially improves network efficiency and resilience against jamming attacks. The findings provide strong evidence that the proposed mitigation approach significantly reduces energy consumption. The changes in variance confirm that energy behavior is altered post-mitigation, and the statistical validation ensures the robustness of the reported improvements. These results strengthen the credibility of the study and confirm that the proposed strategy effectively enhances network performance.

Regarding retransmissions, the normality test evaluates whether the retransmission data follow a normal distribution, where a result of 0 indicates normality and 1 indicates non-normality. In the BLE network, the results show that retransmission data before and after mitigation do not follow a normal distribution, suggesting that the retransmission values are likely skewed or have non-Gaussian characteristics. In contrast, for the LoRa network, the retransmissions were normally distributed before mitigation but deviated from normality after mitigation, indicating a shift in data distribution. The BLE-LoRa Hybrid network exhibits a similar behavior to the BLE network, with both datasets not following a normal distribution before and after mitigation. Levene’s test determines whether the variability (variance) in retransmissions remained consistent before and after mitigation. A high *p*-value suggests that the variance is homogeneous, while a low *p*-value implies significant differences in variance. In the BLE network, the *p*-value of 0.54121 suggests that the retransmission variability remained stable after mitigation. Similarly, the LoRa network’s *p*-value of 0.19191 indicates that its retransmission variance also remained relatively unchanged. However, the BLE-LoRa Hybrid network had a *p*-value of 0.00000, signifying a drastic change in variance, meaning the mitigation strategy significantly altered the retransmission patterns. The statistical significance test examines whether the difference in retransmissions before and after mitigation is statistically significant. A low *p*-value indicates that the difference is meaningful and unlikely due to random variation. The BLE network results show a *p*-value of 0.00000, confirming a highly significant difference in retransmissions after mitigation. The LoRa network also experienced a statistically significant change, with a *p*-value of 0.00205, suggesting that retransmissions were successfully reduced. The BLE-LoRa Hybrid network had a *p*-value of 0.00040, which is also very low, indicating a substantial shift in retransmission behavior after mitigation. The results indicate that the mitigation strategy was effective across all networks, reducing retransmissions significantly. The BLE network maintained stable variance, suggesting that while retransmissions were reduced, the pattern of retransmission values remained similar. In the LoRa network, the shift from normal to non-normal distribution after mitigation suggests a more irregular impact on retransmission values. The BLE-LoRa Hybrid network was the most impacted, showing significant changes in both retransmission counts and variance, indicating a major structural shift in network behavior after mitigation. These findings reinforce the effectiveness of the mitigation strategy, with the BLE-LoRa Hybrid network experiencing the most substantial transformation.

## 4. Results and Discussion

To validate the effectiveness of the cluster-based mitigation algorithm in detecting and counteracting jamming attacks, three different test scenarios are proposed. These scenarios analyze three key mitigation metrics: retransmission rate, energy consumption, and routing resilience. The experiments are conducted in a university campus environment, with sensor nodes deployed in various locations such as classrooms, offices, laboratories, and outdoor areas. Each test scenario introduces a jammer node to simulate a targeted attack while evaluating how the network adapts through clustering mechanisms.

### 4.1. Scenario 1: BLE-Based Communication with Cluster-Based Mitigation and a Jammer Node

In this scenario, the sensor network operates exclusively using BLE, where LPSTK-CC1352R sensors communicate directly with the Laird Sentrius Gateway. A subset of CC1352P sensors is configured as cluster heads based on their energy availability, retransmission rates, and link quality. The jammer node is positioned near one of the CHs to introduce interference, causing packet losses and increased retransmission rates. The goal is to evaluate how the network adapts through cluster reconfiguration, where affected nodes elect a new CH and reroute data through alternative BLE paths.

Key metrics evaluated:Retransmission Rate: The number of packet retransmissions before successful delivery.Energy Consumption: Power usage before and after cluster adaptation.Routing Resilience: The time taken to reestablish a stable BLE communication path after jamming.

In the first phase, sensor nodes operate exclusively in BLE mode without any mitigation strategies. Energy consumption data are collected every hour over 24 h to establish a baseline measurement of power usage under normal network conditions. This behavior is described in Figure 2, Figure 3 and Figure 4. In the second phase, conducted on a separate day, the proposed mitigation algorithm is implemented within the BLE network. The same set of sensor nodes is monitored for energy consumption every hour for another 24-h period. During this phase, the mitigation strategy dynamically adjusts transmission power, reroutes data through alternative paths, and reorganizes network clusters to counteract potential jamming attacks. Comparative analysis between the two datasets allows for the evaluation of the effectiveness of the mitigation strategy in reducing energy consumption while maintaining network stability.

Figure 2 illustrates the energy consumption trends for BLE under jamming conditions, both without and with the mitigation algorithm. The results demonstrate that, in the absence of a mitigation strategy, energy consumption increases progressively as jamming intensifies, primarily due to the elevated retransmission rates caused by packet collisions and interference. Without mitigation, affected nodes continue to attempt transmissions, leading to unnecessary energy expenditure. Conversely, when the mitigation algorithm is applied, energy consumption is significantly reduced. This efficiency gain can be attributed to the algorithm’s adaptive routing approach, which dynamically reorganizes network clusters and redistributes transmission responsibilities to nodes with higher energy reserves, thereby optimizing power usage. Furthermore, the implementation of intelligent transmission power adjustments minimizes excessive retransmissions and improves network stability.

Figure 3 highlights the variation in retransmission rates under similar jamming conditions. The number of retransmissions in the unprotected BLE network remains notably high, particularly during peak interference periods, as the nodes continuously attempt to deliver packets despite heavy congestion in the communication channel. This behavior severely affects overall network efficiency, leading to increased latency and higher energy drainage. However, with the mitigation strategy in place, the retransmission rate drops substantially. This improvement stems from the algorithm’s ability to detect jamming zones in real time and reconfigure communication paths accordingly. By selecting alternative routes or temporarily shifting transmission responsibilities to more robust nodes, the system reduces failed transmission attempts, thus alleviating network congestion.

Figure 4 presents the resilience of the BLE network, measured as the time required for the network to recover after a jamming event. Without mitigation, the recovery time is prolonged due to the network’s inability to promptly adapt to the changing interference landscape. The absence of an intelligent coordination mechanism means that affected nodes persist in unsuccessful transmission attempts, delaying network stabilization. In contrast, the network employing the proposed mitigation algorithm exhibits a considerably lower recovery time. The key factor contributing to this improvement is the dynamic reconfiguration of network clusters, which ensures that disrupted nodes swiftly identify new cluster heads or alternative routing paths, thereby restoring connectivity more efficiently.

### 4.2. Scenario 2: LoRa-Based Communication with Cluster-Based Mitigation and a Jammer Node

In this experiment, the network operates solely on LoRa communication, where CC1352P sensors use LoRaWAN to transmit data to the Laird Sentrius Gateway. Each CH aggregates data from its cluster members and relays them to the gateway via long-range LoRa links. A LoRa jammer is introduced near one CH, causing high interference levels in the sub-GHz spectrum. The mitigation strategy relies on alternative CH selection and rerouting through neighboring CHs.

Key metrics evaluated:Retransmission Rate: The number of attempts needed before successful data reception at the gateway.Energy Consumption: The impact of increased LoRa transmissions due to jamming.Routing Resilience: The time required for affected nodes to reroute their packets through an alternative CH.

The results depicted in Figure 5, Figure 6 and Figure 7 provide a comprehensive evaluation of the impact of jamming on LoRa-based communication and the effectiveness of the proposed cluster-based mitigation algorithm. Figure 5 highlights the energy consumption trend in a LoRa network under jamming conditions, comparing scenarios with and without the mitigation algorithm. In the absence of mitigation, energy consumption steadily increases due to frequent retransmissions caused by interference in the sub-GHz spectrum. This trend demonstrates the inefficiency of conventional LoRa communication when facing external attacks, as nodes continuously attempt to transmit unsuccessfully, leading to excessive power drain. However, with the mitigation algorithm in place, the network dynamically reorganizes itself, redistributing transmission responsibilities among the most resilient nodes and optimizing power usage. This results in a significant reduction in overall energy consumption, proving the efficiency of adaptive cluster selection and routing adjustments in preserving energy resources. Figure 6 presents a comparative analysis of retransmission rates under jamming conditions. In the unmitigated scenario, retransmissions remain persistently high due to continuous interference, leading to packet delivery failures and network congestion. This behavior severely impacts communication reliability, increasing latency and causing unnecessary power expenditure. When the proposed mitigation strategy is implemented, retransmission rates drop substantially as the algorithm promptly identifies affected nodes and reconfigures communication paths accordingly. This intelligent rerouting reduces failed transmission attempts and alleviates congestion, leading to a more stable and efficient network. Finally, Figure 7 illustrates the resilience of the LoRa network, measured by the recovery time required for the network to reestablish stable communication after a jamming attack. Without mitigation, recovery time is significantly prolonged due to the lack of an adaptive response mechanism, leaving affected nodes in a persistent state of communication failure. Conversely, with the mitigation algorithm, network recovery is expedited through proactive cluster reconfiguration and adaptive transmission adjustments. This rapid adaptation ensures that alternative routes are established efficiently, reducing downtime and maintaining network continuity. Collectively, these results underscore the effectiveness of the cluster-based mitigation approach in enhancing the robustness and energy efficiency of LoRa communication in jamming-prone environments. The findings emphasize the critical need for adaptive routing and network reconfiguration mechanisms to counteract the detrimental effects of interference, ensuring resilient long-range wireless communication in real-world IoT deployments.

### 4.3. Scenario 3: Adaptive Communication Switching Between LoRa and BLE with Cluster-Based Mitigation

This scenario tests the adaptive switching mechanism between LoRa and BLE when a jamming attack occurs. Initially, CC1352P and LPSTK-CC1352R sensors communicate using their preferred mode—LoRa for long-range data transmission and BLE for short-range communication. When a jammer disrupts BLE communication, affected nodes automatically switch to LoRa. Conversely, if the LoRa channel is jammed, nodes switch to BLE. The system dynamically selects the most energy-efficient and least-interfered communication path while maintaining clustering.

Key metrics evaluated:Retransmission Rate: The number of failed transmissions before switching to an alternative protocol.Energy Consumption: The additional energy overhead due to protocol switching.Routing Resilience: The time required for the system to detect jamming and transition to the alternative communication mode.

The results presented in Figure 8, Figure 9 and Figure 10 offer a detailed assessment of the energy efficiency, retransmission rates, and resilience of the hybrid BLE-LoRa network under jamming conditions, considering the implementation of the proposed cluster-based mitigation algorithm. Figure 8 illustrates the energy consumption trends when the network operates under dynamic switching between BLE and LoRa. In the absence of the mitigation strategy, energy consumption remains inconsistent and significantly higher due to frequent protocol switching, which occurs inefficiently in response to interference. This behavior suggests that without an intelligent decision-making mechanism, the network expends unnecessary energy while attempting to maintain connectivity. However, when the proposed mitigation algorithm is integrated, energy consumption is optimized as the system adaptively selects the most efficient communication protocol based on real-time network conditions. The algorithm ensures that BLE is prioritized in low-interference environments to capitalize on its low power consumption, while LoRa is utilized strategically for longer-range communication when BLE is severely affected by jamming. This intelligent switching mechanism reduces unnecessary protocol transitions, resulting in overall energy savings. Figure 9 provides insights into retransmission rates, which, in the unmitigated scenario, remain high due to the instability introduced by uncontrolled switching between BLE and LoRa. The lack of a structured transition mechanism causes frequent packet losses, forcing repeated transmission attempts that lead to excessive network congestion. With the implementation of the mitigation algorithm, retransmission rates drop significantly as the system effectively balances communication load across BLE and LoRa, ensuring that packets are routed through the most reliable and least congested paths. The mitigation strategy employs real-time channel assessment and adaptive transmission power adjustments, reducing the need for redundant transmissions and improving network efficiency. Figure 10 evaluates the resilience of the hybrid network, measuring the recovery time required to restore stable communication after a jamming attack. Without mitigation, the recovery time is prolonged due to the network’s reactive rather than proactive approach in dealing with interference, leading to delays in selecting the optimal communication mode. In contrast, the implementation of the mitigation algorithm allows the network to swiftly adjust to changing interference conditions by dynamically reconfiguring its topology and making predictive decisions on protocol switching. This proactive adaptation significantly reduces recovery time, ensuring minimal disruption and improved continuity of communication. Collectively, these findings highlight the effectiveness of the proposed mitigation algorithm in optimizing the hybrid BLE-LoRa network’s performance under jamming conditions. The results validate that by integrating intelligent switching, adaptive routing, and cluster-based coordination, the network not only achieves lower energy consumption and reduced retransmissions but also exhibits enhanced resilience against interference. This underscores the necessity of advanced network management strategies to fully harness the benefits of hybrid communication systems in real-world IoT applications where reliability and energy efficiency are critical.

### 4.4. Comparative Energy Consumption Analysis

Another analysis provides two critical visualizations that allow for a deep understanding of energy consumption across three network configurations: BLE (Bluetooth Low Energy), LoRa (long-range communication), and BLE+LoRa (hybrid switching between BLE and LoRa). These visualizations not only facilitate direct comparisons but also highlight underlying statistical properties that impact the efficiency and variability of each communication protocol.

From the boxplots, we can assess which network configuration exhibits higher energy efficiency by comparing the median values. If one configuration consistently has a lower median energy consumption, it implies a more energy-efficient operation. Additionally, the spread of the box and whiskers indicates the consistency of the energy consumption. A wider IQR or longer whiskers suggests higher variability, which may indicate unpredictable energy consumption patterns, potentially affecting network stability. If a configuration has numerous outliers, it signals instances where energy consumption deviates significantly from the expected range, which could be problematic in resource-constrained environments. Comparing BLE, LoRa, and BLE+LoRa, we can critically evaluate trade-offs between energy efficiency and network performance. BLE, typically designed for short-range and low-energy applications, is expected to have lower energy consumption but may suffer from limited coverage. On the other hand, LoRa, optimized for long-range communication, likely shows higher energy consumption but ensures greater reach and coverage. The BLE+LoRa hybrid approach aims to balance these trade-offs, adapting between protocols based on network conditions. If the boxplot reveals that BLE+LoRa has a median closer to BLE but a higher spread, it suggests that the hybrid approach achieves some energy savings but introduces variability due to switching.

The boxplot analysis presented in Figure 11 provides insights into the energy consumption trends of BLE, LoRa, and the hybrid BLE-LoRa switching mechanism under jamming conditions, both with and without the proposed mitigation algorithm. The results reveal a clear distinction in energy usage across the three network configurations, with BLE exhibiting the lowest median energy consumption due to its inherently low-power communication model but suffering from significant variability when exposed to interference. LoRa, while designed for long-range transmission, demonstrates significantly higher energy usage due to its prolonged transmission durations and greater power requirements. However, the most notable observation emerges from the BLE-LoRa switching mechanism, which, in the absence of mitigation, exhibits erratic energy consumption due to inefficient protocol transitions triggered by interference.

The experimental setup that produced these specific distributions consisted of a series of controlled tests conducted in a university campus environment. Sensor nodes were deployed in various locations, including laboratories, classrooms, and open areas, ensuring diverse interference conditions. Data were collected over multiple experimental runs, capturing energy consumption at regular intervals while the network operated under jamming attacks. The mitigation algorithm played a crucial role in stabilizing the hybrid BLE-LoRa approach by dynamically selecting the most energy-efficient communication protocol based on real-time network conditions. By ensuring that BLE was prioritized in low-interference environments and LoRa was used selectively for resilient communication, the algorithm minimized unnecessary protocol switching and reduced overall energy consumption.

The results validate that an adaptive, intelligent switching mechanism significantly improves energy efficiency in heterogeneous networks, ensuring optimal performance in jamming-prone environments. The implementation of the mitigation algorithm stabilized the hybrid approach by dynamically managing energy expenditure, mitigating inefficiencies in protocol transitions, and reducing fluctuations caused by jamming-induced overhead. These findings emphasize the importance of intelligent protocol management in maintaining energy efficiency and resilience in wireless sensor networks under interference conditions.

### 4.5. Histogram with Probability Density Function

From this analysis, we can determine whether the energy consumption follows a normal distribution, is skewed towards lower or higher values, or exhibits multiple peaks. A narrow and tall peak suggests that most energy values cluster around a specific range, indicating a consistent and predictable energy consumption pattern. In contrast, a wider or multi-modal distribution suggests fluctuations and inconsistencies in energy usage. If BLE exhibits a tight peak at lower values, it confirms that it is the most energy-efficient. If LoRa shows a wider and flatter distribution, it suggests more variability in energy demand, which could stem from varying transmission distances. The BLE+LoRa configuration, ideally, should demonstrate a balanced peak that does not lean too much towards either extreme, indicating an adaptive trade-off between efficiency and performance.

Furthermore, the histogram helps in identifying potential inefficiencies. If a network protocol has a long tail extending toward higher energy values, it implies occasional spikes in energy consumption, which could lead to battery drain issues in IoT devices. The presence of these spikes in the BLE+LoRa configuration would indicate that switching between protocols sometimes results in suboptimal energy efficiency, depending on network conditions.

Figure 12 presents the energy consumption distribution through a histogram and probability density function (PDF), highlighting the statistical characteristics of each communication mode under jamming conditions. BLE exhibits a narrow and sharply peaked distribution, reinforcing its status as the most energy-efficient protocol, albeit with noticeable sensitivity to interference, which causes occasional spikes in energy usage. LoRa, on the other hand, has a broader distribution, with a long rightward tail, indicating inconsistent but generally higher energy consumption, particularly during retransmissions caused by jamming. The BLE-LoRa switching mechanism displays a uniform distribution, with a peak aligning more closely with BLE, suggesting improved efficiency and reduced unnecessary protocol transitions. This finding underscores the importance of adaptive network management in hybrid communication systems, demonstrating that intelligent switching mechanisms can achieve a balance between energy efficiency and network resilience. These insights emphasize the necessity of real-time decision making in protocol selection to minimize energy wastage and enhance the sustainability of wireless sensor networks in challenging environments.

Figure 12 was constructed based on experimental data collected from real-world jamming scenarios in a heterogeneous LoRa-BLE wireless sensor network. The dataset consists of the energy network performance metric recorded over multiple controlled experiments conducted within a university campus environment. The data were sampled at regular intervals, ensuring uniform time distribution across different interference conditions. Each recorded data point corresponds to a measurement taken from sensor nodes deployed in varying locations, including laboratories, classrooms, and open areas, where interference sources such as Wi-Fi networks and mobile devices were present. To generate the histograms, we used a binning strategy that appropriately reflects the data distribution without introducing bias or excessive granularity. The number of bins was determined using the Freedman–Diaconis rule, which balances the need for detailed resolution with statistical significance. The sampling methodology ensured data representativeness by incorporating different interference intensities and network conditions. We performed preprocessing steps such as outlier detection and removal using interquartile range (IQR) filtering to eliminate potential anomalies that could distort the statistical representation. Moreover, normality tests (Shapiro–Wilk and Anderson–Darling) were conducted to determine whether the datasets followed Gaussian distributions, guiding the selection of appropriate statistical analyses. By applying these methods, the histograms and PDFs accurately illustrate the statistical behavior of the network under jamming conditions before and after the mitigation strategy. The results offer valuable insights into how energy consumption was affected by interference and how the proposed mitigation mechanism effectively altered the network’s statistical characteristics.

### 4.6. Potential Limitations

The proposed mitigation strategy was primarily evaluated under reactive and constant jamming conditions, which are among the most common threats in WSNs. However, other sophisticated forms of jamming, such as random and deceptive jamming, could pose additional challenges to network resilience. Random jamming, characterized by alternating interference periods, may result in intermittent network disruptions that could delay the cluster-based adaptation process. Similarly, deceptive jamming, where malicious packets are injected to confuse network protocols, might undermine routing decisions and lead to inefficient energy utilization. While our current strategy relies on adaptive cluster selection and energy-aware protocol switching, its effectiveness against these advanced jamming tactics remains an open research question. Furthermore, the impact of adversarial machine learning techniques on jamming strategies is an emerging threat. Attackers could potentially use reinforcement learning to optimize jamming patterns, making them more difficult to detect and mitigate. Although our algorithm dynamically reconfigures network topology based on real-time performance metrics, future work should explore how intelligent adversarial attacks could adapt to and counteract mitigation efforts. In particular, the integration of AI-driven anomaly detection and predictive analytics could enhance the robustness of our approach by proactively identifying evolving interference patterns and adjusting mitigation strategies accordingly.

Another potential limitation lies in the scalability of the mitigation strategy. While our experiments were conducted in a controlled university campus environment, large-scale deployments in industrial IoT or smart city applications might face additional challenges. The increased number of nodes and varying interference sources could lead to higher computational overhead for cluster selection and protocol switching, potentially affecting response times. Future research should investigate distributed decision-making mechanisms that allow for real-time adaptation with minimal processing requirements.

These considerations underscore the need for further studies to validate the proposed mitigation framework under diverse jamming conditions and larger-scale network deployments. By addressing these aspects, we can enhance the adaptability and robustness of our approach, ensuring its effectiveness across a broader range of real-world scenarios.

## 5. Conclusions

The findings of this study highlight the substantial impact of the proposed jamming mitigation algorithm on network resilience, energy efficiency, and communication stability within heterogeneous WSNs. By implementing an adaptive cluster-based mitigation strategy, the system successfully counteracts jamming attacks in BLE, LoRa, and hybrid BLE-LoRa networks, ensuring reliable data transmission even in interference-prone environments. The results demonstrate that, in BLE-only configurations, the mitigation algorithm significantly reduces retransmission rates and energy consumption by dynamically reorganizing cluster heads and optimizing transmission power. Similarly, in LoRa-based networks, where energy usage is inherently higher due to longer transmission distances, the proposed strategy mitigates energy drain by intelligently rerouting traffic and selecting alternative communication paths. The hybrid BLE-LoRa approach, leveraging the strengths of both technologies, showcases the highest adaptability under jamming conditions, effectively balancing energy efficiency and transmission range through real-time protocol switching. The comparative analysis validates that intelligent network adaptation not only enhances the robustness of WSNs against external interference but also extends the operational lifetime of battery-powered devices, making it a crucial advancement for large-scale IoT deployments.

The core contribution of this research lies in its novel integration of a cluster-based jamming mitigation algorithm with multi-protocol network switching, setting it apart from conventional static anti-jamming strategies. Unlike prior approaches that rely on frequency hopping or power adaptation alone, the proposed method dynamically adjusts network topology and transmission parameters based on real-time interference detection. This innovation ensures that affected nodes rapidly adapt to changing conditions, reducing packet loss and minimizing energy-intensive retransmissions. Furthermore, the study provides a comprehensive empirical evaluation of the trade-offs between BLE, LoRa, and hybrid BLE-LoRa configurations, offering practical insights into their energy-performance dynamics under jamming scenarios. The results not only reinforce the effectiveness of adaptive mitigation strategies in heterogeneous IoT networks but also establish a foundation for future research on AI-driven jamming resilience mechanisms. By integrating intelligent decision making with multi-protocol flexibility, this work paves the way for more secure and energy-efficient WSN architectures, addressing critical challenges in wireless communication for smart cities, industrial automation, and large-scale sensor deployments.

While our proposed mitigation strategy significantly improves resilience against jamming attacks in HWSNs, several challenges remain. One major area for future work is the extension of our algorithm to address more sophisticated and adaptive jamming techniques, such as adversarial machine-learning-based jammers that dynamically alter their interference patterns. Additionally, although our approach effectively balances energy efficiency and interference mitigation, further research is needed to optimize protocol switching decisions under varying network loads and mobility conditions. Another potential direction is the exploration of collaborative mitigation strategies that integrate multiple nodes’ intelligence to detect and counteract jamming at a network-wide level rather than relying solely on cluster-based adaptation. Finally, while our experimental setup provides valuable real-world validation, large-scale deployments in diverse environments, such as industrial IoT and smart city applications, could offer deeper insights into scalability and long-term network performance under prolonged jamming scenarios. By considering these aspects, future studies can build upon our work to develop more robust and adaptive solutions for securing HWSNs against evolving interference threats.

## Figures and Tables

**Figure 1 sensors-25-01931-f001:**
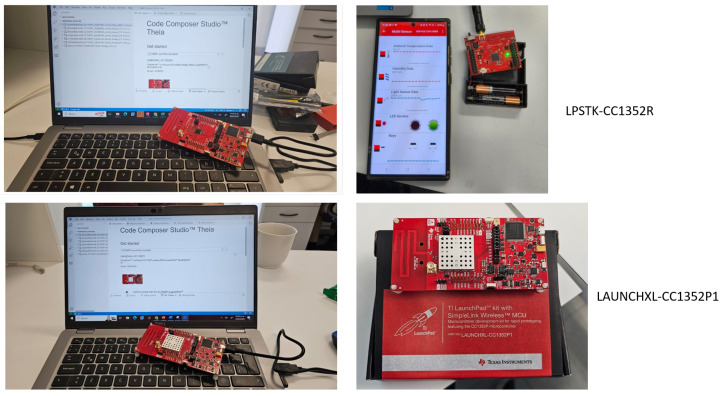
Implementation of the sensor network.

**Figure 2 sensors-25-01931-f002:**
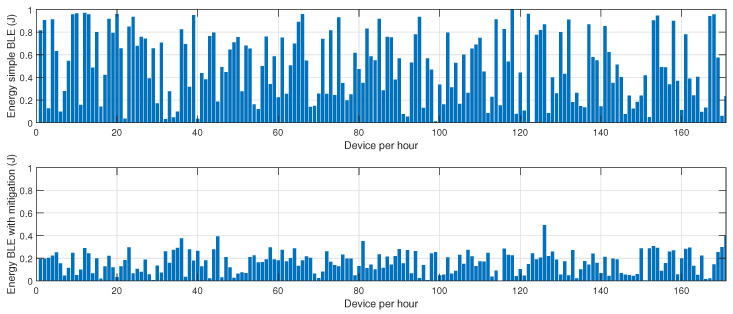
Impact of the proposed mitigation algorithm on BLE energy consumption under jamming conditions.

**Figure 3 sensors-25-01931-f003:**
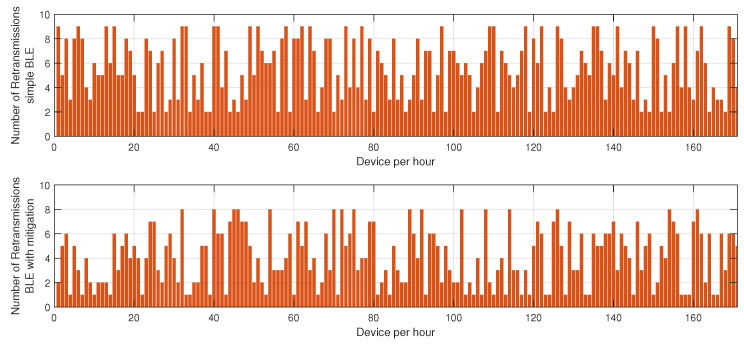
Impact of the proposed mitigation algorithm on BLE retransmissions under jamming conditions.

**Figure 4 sensors-25-01931-f004:**
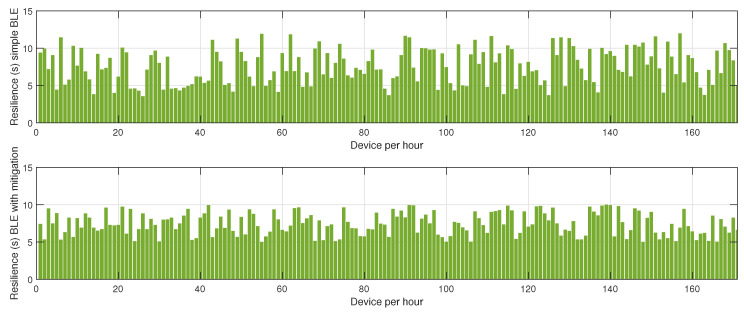
Impact of the proposed mitigation algorithm on BLE resilience under jamming conditions.

**Figure 5 sensors-25-01931-f005:**
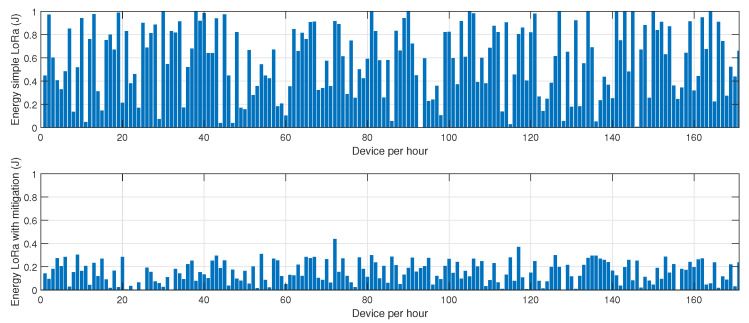
Impact of the proposed mitigation algorithm on LoRa energy consumption under jamming conditions.

**Figure 6 sensors-25-01931-f006:**
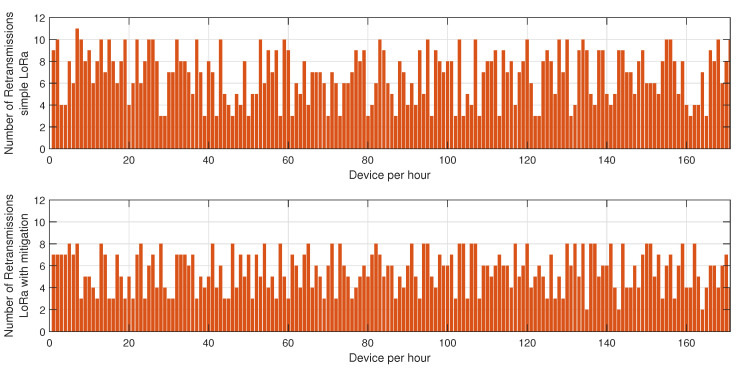
Impact of the proposed mitigation algorithm on LoRa retransmissions under jamming conditions.

**Figure 7 sensors-25-01931-f007:**
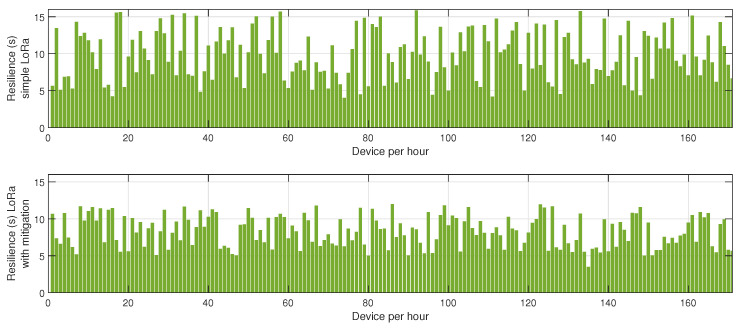
Impact of the proposed mitigation algorithm on LoRa resilience under jamming conditions.

**Figure 8 sensors-25-01931-f008:**
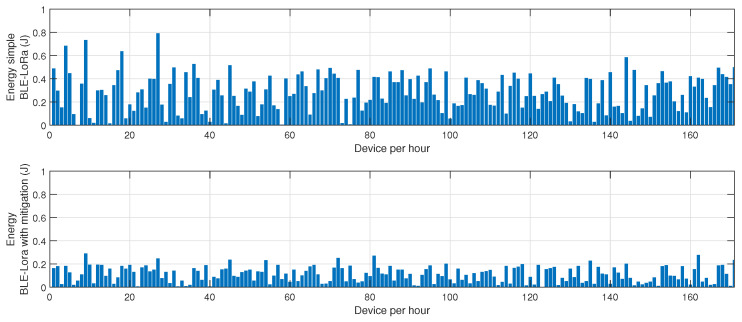
Impact of the proposed mitigation algorithm on LoRa and BLE energy consumption under jamming conditions.

**Figure 9 sensors-25-01931-f009:**
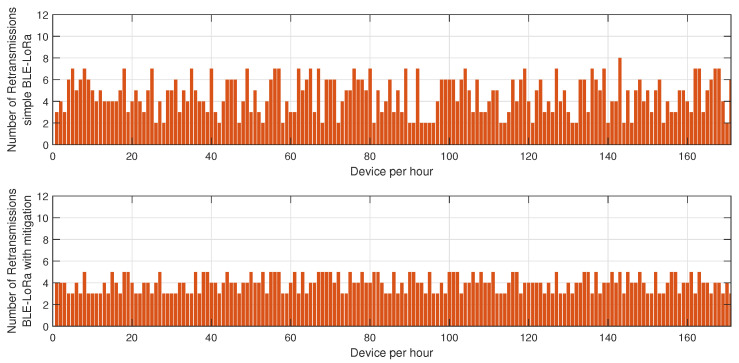
Impact of the proposed mitigation algorithm on LoRa and BLE retransmissions under jamming conditions.

**Figure 10 sensors-25-01931-f010:**
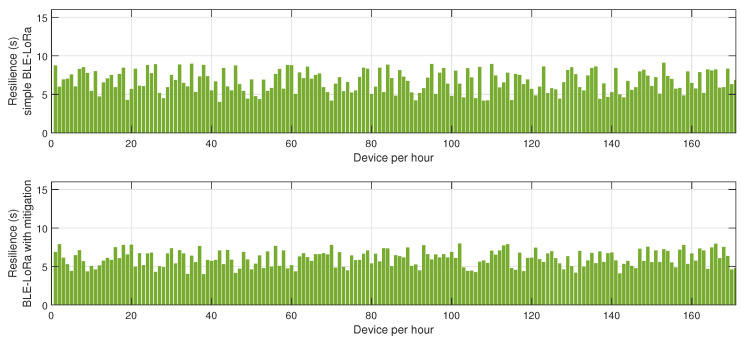
Impact of the proposed mitigation algorithm on LoRa and BLE resilience under jamming conditions.

**Figure 11 sensors-25-01931-f011:**
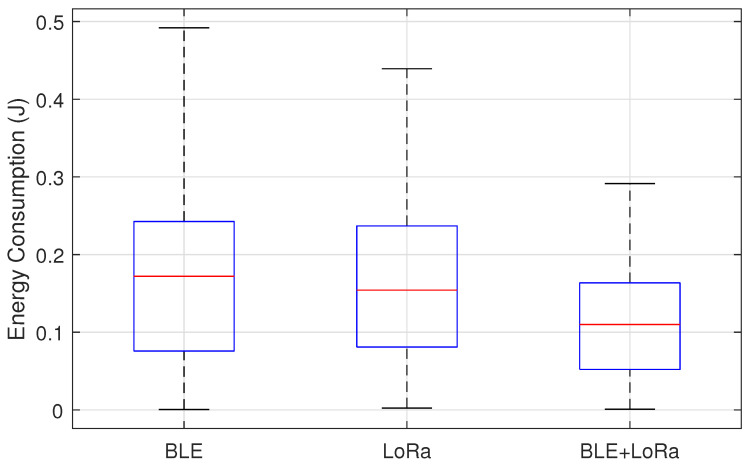
Energy consumption analysis for BLE, LoRa, and hybrid BLE-LoRa switching under jamming conditions with and without mitigation.

**Figure 12 sensors-25-01931-f012:**
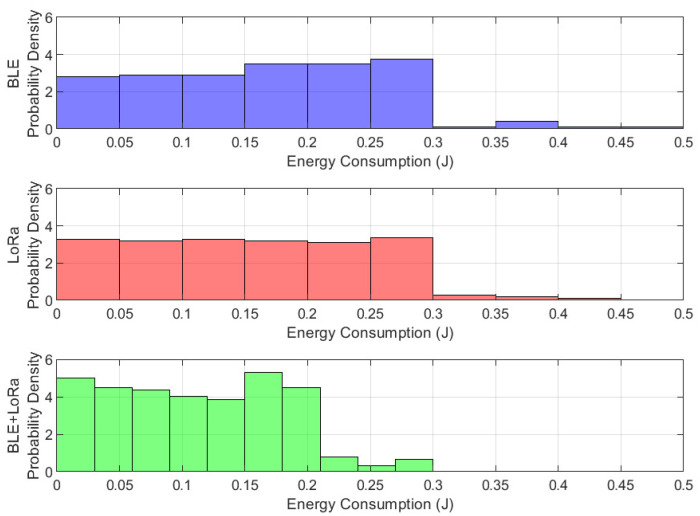
Histogram and probability density function of energy consumption for BLE, LoRa, and BLE-LoRa under jamming conditions.

**Table 1 sensors-25-01931-t001:** Current references on wireless sensor networks and their applications for elderly health monitoring and behavioral change techniques.

Reference	Energy Consumption Method	Wireless Technologies	Main Contribution	Limitations
Jaber et al. (2020) [28]	Adaptive duty cycling	BLE, Wi-Fi	Energy-efficient protocol switching based on interference	High latency during high interference periods
Popli et al. (2018) [29]	Sleep scheduling	LoRa, NB-IoT	Optimized sleep-wake scheduling to reduce energy waste	Inefficient in highly dynamic environments
Shaabanzadeh et al. (2024) [30]	Machine learning-based power control	5G, Wi-Fi	Intelligent power adaptation based on traffic prediction	Requires high computational resources
Gupta et al. (2022) [31]	Energy-aware clustering	LoRa, Zigbee	Cluster-based transmission to minimize redundant messages	Not scalable for dense networks
Mohanty (2010) [32]	Power-aware routing	BLE, Zigbee	Dynamic path selection for energy efficiency	Limited for long-range communication
Nelson et al. (2023) [33]	Hybrid network switching	BLE, LoRa, 5G	Adaptive switching between BLE and LoRa for efficiency	Increased network overhead
Alves et al. (2021) [34]	RF energy harvesting	LoRa, NB-IoT	Utilizes ambient RF energy to extend battery life	Requires additional RF energy sources
Park et al. (2020) [35]	Transmission power control	Zigbee, BLE	Adjusts power levels dynamically based on network load	Can cause instability in low-power nodes
Hussain et al. (2020) [36]	AI-based resource allocation	5G, LoRa	Uses reinforcement learning for energy efficiency	Requires extensive training data
Olatinwo et al. (2021) [37]	Energy-efficient MAC protocols	LoRaWAN, Wi-Fi	Reduces idle listening time for better power savings	Increased delay in high-traffic scenarios
Abderrahmane et al. (2024) [38]	Low-power adaptive communication	BLE, Zigbee, LoRa	Implements an adaptive protocol selection based on conditions	Overhead increases with multiple nodes
Teixeira (2022) [39]	Opportunistic data transmission	5G, Wi-Fi	Reduces power usage by leveraging network conditions	Not optimized for real-time applications
Poyyamozhi et al. (2024) [40]	AI-driven power optimization	LoRa, Zigbee	Predictive analytics for reducing energy drain	Complex deployment in large networks
Abadi et al. (2022) [41]	Dynamic frequency hopping	NB-IoT, Wi-Fi	Avoids interference by switching frequencies adaptively	Increased complexity in synchronization
Nandal et al. (2021) [42]	Cluster-based data aggregation	Zigbee, LoRa	Minimizes redundant data transmission to save energy	Performance drops with high mobility nodes
Aldhaheri et al. (2024) [43]	Context-aware adaptive transmission	LoRa, BLE	Optimizes energy consumption based on environmental changes	Requires precise calibration for effectiveness
Alselek et al. (2023) [44]	AI-driven dynamic power scaling	5G, LoRa	Reduces power usage through real-time AI predictions	High computational overhead for AI model training
Paul et al. (2024) [45]	Multi-hop energy-efficient routing	Zigbee, BLE	Increases network lifetime by optimizing multi-hop paths	Limited performance in dense deployments
Ntabeni et al. (2024) [46]	Hybrid energy-aware MAC protocol	Wi-Fi, NB-IoT	Reduces idle listening time and improves power savings	Increased complexity in synchronization

## Data Availability

Data are contained within the article and Appendix A.

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
