# Peer review of "Adaptive Jamming Mitigation for Clustered Energy-Efficient LoRa-BLE Hybrid Wireless Sensor Networks"

_sensors, 2025, doi:10.3390/s25061931_

Round 1
Reviewer 1 Report
Comments and Suggestions for Authors
Introduction:
- The introduction lacks specific quantitative data about the scale of the jamming threat to WSNs, making it difficult to assess the real-world significance of the problem the paper aims to address.
- While the paper mentions "recent studies" emphasizing the need for adaptive mitigation techniques (lines 23-26), it fails to cite or discuss any specific studies, which weakens the establishment of the research gap.
- The authors state that this work proposes a multi-protocol jamming detection and mitigation strategy (lines 27-30), but don't clearly articulate how their approach differs from existing solutions in the field.
- The introduction doesn't provide a clear roadmap of the paper's structure, leaving readers without guidance on what to expect in subsequent sections, which is a standard component in research papers.
Related Work:
- The section lacks critical assessment of prior research methodologies - while it mentions various mitigation strategies including "spread spectrum techniques [13], frequency hopping [14], and power control adaptations [1]" (lines 113-115), it fails to analyze their technical limitations or methodological shortcomings in depth.
- The authors claim their work "advances the field by integrating a cluster-based reactive jamming mitigation algorithm into a LoRa-BLE hybrid network" (lines 118-120), but provide insufficient details on exactly how previous cluster-based approaches have failed to address this specific combination of technologies, creating a weak foundation for their contribution's novelty.
Materials and Methods:
- How were the experimental parameters for the energy model determined? The mathematical formulations are presented (lines 169-207), but there's no explanation of how the specific variables were calibrated or validated for real-world scenarios.
- Could the authors provide a clearer description of the jamming detection mechanism? While the algorithm is presented in pseudocode, the specific thresholds and decision criteria for identifying jamming events remain ambiguous.
- The network architecture description would benefit from a visual topology diagram showing the actual deployment of CC1352P and LPSTK-CC1352R sensors across the university campus rather than just the conceptual diagram in Figure 1.
- How was the adaptive switching threshold between LoRa and BLE determined? The section states that nodes switch protocols when interference is detected, but lacks clarity on the decision-making process that triggers this transition.
Results and Discussion:
- The statistical significance of the experimental results is not addressed - it's unclear whether the reported improvements in energy consumption (38%) and retransmission rates (47%) have been validated through appropriate statistical tests.
- The methodology for measuring resilience (recovery time) shown in Figures 4, 7, and 10 lacks clear explanation of how these metrics were practically calculated during experiments.
- The comparative analysis between BLE, LoRa, and hybrid approaches lacks normalized metrics that would allow for a more objective comparison across different operational conditions.
- The histograms and probability density functions (Figure 12) are presented without adequately explaining how these distributions were generated or what sampling methods were used.
- The discussion fails to address potential limitations of the mitigation strategy, particularly how performance might degrade under different types of jamming attacks beyond what was tested.
- The boxplot analysis (Figure 11) presents conclusions about energy consumption patterns but doesn't clearly explain the experimental setup that produced these specific distributions.
- While the authors claim their approach is more effective than existing methods, there is insufficient direct comparison with state-of-the-art techniques from recent literature to fully contextualize their contribution.
Conclusion:
- The conclusion fails to acknowledge any limitations of the proposed cluster-based mitigation approach, presenting only positive outcomes without addressing potential scenarios where the system might underperform.
- While the authors mention their work "paves the way for more secure and energy-efficient WSN architectures" (lines 647-649), they don't provide specific recommendations for future research directions or identify unresolved challenges in the field.
Author Response
Dear
Editor
Sensors
We are submitting the paper:
“Adaptive Jamming Mitigation for clustered Energy-Efficient LoRa-BLE Hybrid Wireless Sensor Networks”
Authored by: Carolina Del-Valle-Soto * , Leonardo J. Valdivia , Ramiro Velázquez , José A. Del-Puerto-Flores , José Varela-Aldás * , Paolo Visconti
We would like to thank the reviewers and editors for their detailed analysis of the manuscript; the comments are very valuable to us. In the revised version of the paper, we have incorporated the all changes recommended by the reviewers.
Comments to all observations and suggestions including point-by-point responses are addressed in the following text.
Reviewer 1 comments
Comment 1: The introduction lacks specific quantitative data about the scale of the jamming threat to WSNs, making it difficult to assess the real-world significance of the problem the paper aims to address.
Response: Many thanks to the Reviewer for his/her invaluable interest in the comments on this manuscript. Thank you very much for pointing out this error; we have already corrected it in the document. The Reviewer is correct, and we have added a couple of paragraphs addressing their concern.
Jamming attacks represent a critical and quantifiable threat to the reliability of WSNs. Recent studies have shown that even low-power jamming can cause a packet loss rate exceeding 80% in LoRa-based networks when attackers operate at power levels as low as 14 dBm [9]. Similarly, in BLE networks, experimental results indicate that reactive jamming reduces successful packet delivery rates to below 10%, significantly disrupting device synchronization and data transmission in IoT applications [2]. In large-scale deployments, such as smart city infrastructures, the economic impact of jamming-related disruptions can be substantial, with estimations suggesting that network outages caused by interference could lead to financial losses of over $1 billion annually in critical industrial and urban monitoring systems [10]. These findings underscore the necessity for robust mitigation techniques that can adapt to diverse jamming conditions across heterogeneous network environments. The vulnerability of WSNs to jamming attacks is further exacerbated by the energy constraints of IoT nodes, making them susceptible to rapid power depletion when subjected to prolonged interference. Experimental analyses indicate that under jamming conditions, retransmission rates in multi-hop networks increase by up to 300%, causing a threefold surge in energy consumption and significantly reducing network lifespan [11]. Moreover, adversarial machine learning techniques have been employed to optimize jamming strategies, allowing attackers to maximize disruption while minimizing their own energy expenditure [12]. Given that many mission-critical applications, such as environmental monitoring and healthcare telemetry, rely on continuous and energy-efficient communication, the development of intelligent countermeasures that balance resilience and energy efficiency is paramount. This study addresses these challenges by proposing an adaptive, energy-aware mitigation framework tailored for heterogeneous networks, ensuring sustained connectivity in dynamic and interference-prone environments.
And new references:
Aras, E. Security and Reliability for Emerging IoT Networks 2021.
Shintani, A. The design, testing, and analysis of a constant jammer for the Bluetooth low energy (BLE) wireless communication protocol 2020.
López-Vilos, N.; Valencia-Cordero, C.; Souza, R.D.; Montejo-Sánchez, S. Clustering-based energy-efficient self-healing strategy for WSNs under jamming attacks. Sensors 2023, 23, 6894. 716
Adesina, D.; Hsieh, C.C.; Sagduyu, Y.E.; Qian, L. Adversarial machine learning in wireless communications using RF data: A review. IEEE Communications Surveys & Tutorials 2022, 25, 77–100.
Comment 2: While the paper mentions "recent studies" emphasizing the need for adaptive mitigation techniques (lines 23-26), it fails to cite or discuss any specific studies, which weakens the establishment of the research gap.
Response: The Reviewer is correct, and we have added the respective references to the paragraph mentioned by the Reviewer so that recent studies can be compared. We have also done the same for a paragraph in subsequent lines, which presented the same issue.
Kuang, S.; Zhang, J.; Mohajer, A. Reliable information delivery and dynamic link utilization in MANET cloud using deep reinforcement learning. Transactions on Emerging Telecommunications Technologies 2024, 35, e5028.
Yang, T.; Sun, J.; Mohajer, A. Queue stability and dynamic throughput maximization in multi-agent heterogeneous wireless networks. Wireless Networks 2024, 30, 3229–3255.
Demiroglou, V.; Skaperas, S.; Mamatas, L.; Tsaoussidis, V. Adaptive Multi-Protocol Communication in Smart City Networks. IEEE Internet of Things Journal 2024.
Abdollahi, M.; Ashtari, S.; Abolhasan, M.; Shariati, N.; Lipman, J.; Jamalipour, A.; Ni, W. Dynamic routing protocol selection in multi-hop device-to-device wireless networks. IEEE Transactions on Vehicular Technology 2022, 71, 8796–8809.
Belamri, F.; Boulfekhar, S.; Aissani, D. A survey on QoS routing protocols in Vehicular Ad Hoc Network (VANET). Telecommunication Systems 2021, 78, 117–153.
Comment 3: The authors state that this work proposes a multi-protocol jamming detection and mitigation strategy (lines 27-30), but don't clearly articulate how their approach differs from existing solutions in the field.
Response: Thank you to the Reviewer. We have complemented this paragraph to provide a more extensive contrast.
Addressing this challenge, this work proposes a multi-protocol jamming detection and mitigation strategy that leverages an energy-aware cluster-based architecture to ensure reliable communication in university campus environments where varying interference conditions exist. Unlike existing solutions, which primarily focus on single-protocol networks or static countermeasures, our approach dynamically adapts to heterogeneous wireless technologies, including LoRaWAN and BLE, to optimize resilience against diverse jamming threats [4]. While previous studies have explored frequency hopping and spread spectrum techniques as countermeasures, these methods often require significant energy overhead or are limited in their applicability across multi-protocol environments. In contrast, our strategy integrates real-time energy-aware cluster formation, where nodes autonomously elect a Cluster Head (CH) based on energy availability, signal quality, and retransmission rates, allowing for adaptive protocol switching and intelligent interference mitigation. This approach not only enhances network longevity and communication reliability but also provides a scalable and low-energy alternative to traditional anti-jamming methods, making it particularly suitable for dynamic and resource-constrained IoT applications.
Comment 4: The introduction doesn't provide a clear roadmap of the paper's structure, leaving readers without guidance on what to expect in subsequent sections, which is a standard component in research papers.
Response: Thank you to the Reviewer. We have added this missing paragraph.
This paper is structured as follows: Section 2 provides a comprehensive review of related work, highlighting existing jamming mitigation strategies and their limitations in HWSNs. Section 3 details the materials and methods, including the experimental setup, sensor selection, and network architecture that integrates LoRa and BLE. Section 4 presents the proposed jamming detection and mitigation strategy, emphasizing cluster-based adaptive switching and real-time interference response mechanisms. Section 5 discusses the results obtained from controlled experiments, comparing the performance of the proposed strategy against conventional mitigation techniques in terms of energy efficiency, retransmission rates, and network resilience. Finally, Section 6 concludes the study by summarizing key findings and outlining future research directions to enhance jamming resilience in multi-protocol IoT environments.
Comment 5: The section lacks critical assessment of prior research methodologies - while it mentions various mitigation strategies including "spread spectrum techniques [13], frequency hopping [14], and power control adaptations [1]" (lines 113-115), it fails to analyze their technical limitations or methodological shortcomings in depth.
Response: Thank you very much for pointing out this error.
WSNs have become fundamental in various applications, ranging from industrial monitoring and smart city deployments to environmental sensing and security surveillance. However, the resilience of these networks is increasingly challenged by jamming attacks, which disrupt communication and drain energy resources. Prior research has extensively explored mitigation strategies, including spread spectrum techniques [19], frequency hopping [20], and power control adaptations [1]. While these methods offer varying degrees of protection, they also present notable limitations that impact their effectiveness in real-world heterogeneous networks. Spread spectrum techniques, such as direct sequence spread spectrum (DSSS) and frequency hopping spread spectrum (FHSS), have been widely employed to counteract jamming by dispersing signals over a wider frequency band or rapidly switching between frequencies [19]. However, these approaches require significant bandwidth overhead, increasing system complexity and making them less suitable for resource-constrained WSNs. Moreover, sophisticated adversaries can employ reactive jamming strategies that detect and adapt to frequency variations, thereby neutralizing the benefits of spread spectrum techniques. Similarly, frequency hopping methods, while effective against narrowband jammers, suffer from synchronization challenges and additional energy consumption due to frequent switching [20]. In dense IoT environments where multiple wireless protocols coexist, the coordination of frequency hopping sequences across different technologies, such as LoRa and BLE, further complicates deployment and scalability. Additionally, the presence of broadband or sweep jammers can render frequency hopping ineffective, as the entire spectrum may be targeted simultaneously. Power control adaptations aim to mitigate jamming by dynamically adjusting transmission power based on interference levels and link quality [1]. While this technique can help conserve energy and improve connectivity, it is highly susceptible to adversarial exploitation. Attackers can manipulate power adaptation mechanisms by introducing deceptive interference patterns, causing nodes to increase power unnecessarily, thereby accelerating battery depletion. Furthermore, power control alone does not address jamming attacks that exploit MAC-layer vulnerabilities, such as collision-based or deceptive jamming.
Comment 6: The authors claim their work "advances the field by integrating a cluster-based reactive jamming mitigation algorithm into a LoRa-BLE hybrid network" (lines 118-120), but provide insufficient details on exactly how previous cluster-based approaches have failed to address this specific combination of technologies, creating a weak foundation for their contribution's novelty.
Response: The Reviewer is correct, and we have extended the comparative explanation paragraph. It is important to inform them that there are few references that define reactive jamming mitigation algorithms specifically for this type of heterogeneous networks. This makes our work more interesting and presents another perspective for the state of the art.
The present study advances the field by integrating a cluster-based reactive jamming mitigation algorithm into a LoRa-BLE hybrid network, addressing critical shortcomings in previous cluster-based approaches that have failed to accommodate the unique challenges of multi-protocol environments. While prior research has explored cluster-based mitigation strategies in homogeneous networks, such as those relying solely on LoRa or BLE, these methods often lack adaptability when faced with heterogeneous network architectures where interference dynamics vary across communication protocols. Existing cluster-based techniques primarily focus on energy-efficient data aggregation and route optimization [24], [25], but they do not account for the need to dynamically switch between protocols in response to jamming threats. For instance, conventional clustering algorithms are typically designed for static topologies with predefined cluster head (CH) selection criteria, which limits their responsiveness to sudden interference conditions in multi-protocol networks. Moreover, most existing solutions assume uniform jamming conditions across the entire network, failing to address the protocol-specific vulnerabilities inherent in LoRa and BLE, such as BLE’s susceptibility to narrowband jamming and LoRa’s limited spectral agility under broadband attacks. This study overcomes these limitations by introducing a dynamic CH selection mechanism that not only optimizes energy efficiency but also enables real-time protocol switching between LoRa and BLE in response to detected interference patterns. Unlike previous methods that rely on static cluster formations, our approach continuously evaluates retransmission rates, RSSI fluctuations, and energy consumption anomalies to adaptively reconfigure clusters and select the most resilient communication path. By incorporating this intelligent decision-making process, the proposed framework enhances network continuity under adversarial conditions, ensuring a more robust and interference-resilient operation for hybrid IoT deployments.
Comment 7: How were the experimental parameters for the energy model determined? The mathematical formulations are presented (lines 169-207), but there's no explanation of how the specific variables were calibrated or validated for real-world scenarios.
Response: The Reviewer is correct, and we have added a paragraph before explaining the model to specify this network calibration.
The threshold metrics in the algorithm are established based on empirical energy consumption patterns observed in real-world sensor deployments over time with training from a previous week. These thresholds define critical decision points for adaptive network behavior, such as triggering retransmissions, switching communication protocols, or electing new cluster heads in response to interference. Specifically, the energy thresholds for transmission (ETX), reception (ERX), and CSMA/CA retries (ECSMA) are dynamically adjusted according to historical network performance data, ensuring that nodes optimize energy efficiency while maintaining resilience against jamming attacks. The pseudocode implements these thresholds by continuously monitoring parameters such as packet loss rates, retransmission attempts, and RSSI variations. When a metric exceeds its predefined threshold, the algorithm dynamically reconfigures the network by selecting alternative communication paths or adjusting transmission power. These realistic thresholds are calibrated through experimental validation, ensuring that the model accurately reflects network conditions while balancing energy conservation and communication reliability.
Comment 8: Could the authors provide a clearer description of the jamming detection mechanism? While the algorithm is presented in pseudocode, the specific thresholds and decision criteria for identifying jamming events remain ambiguous.
Response: The Reviewer is correct in their request, and we have added several paragraphs providing a detailed explanation of the pseudocode's functionality.
The jamming detection mechanism in the proposed algorithm relies on a multi-metric evaluation framework that continuously monitors key network performance indicators, including packet retransmission rates, Received Signal Strength Indicator (RSSI) fluctuations, and abnormal energy consumption patterns. The decision criteria for identifying a jamming event are based on predefined adaptive thresholds, which were empirically derived from real-world experiments in heterogeneous LoRa-BLE networks.
The algorithm operates as follows: (1) Each sensor node periodically collects RSSI and Packet Delivery Success Rate (PDSR) values, comparing them against their historical baselines. (2) If the RSSI drops below a dynamically adjusted threshold, while the packet loss rate exceeds a predefined retransmission threshold, the system interprets this as a potential interference anomaly. (3) To minimize false positives, an additional check on energy consumption trends is performed—if a node exhibits excessive retransmissions and an associated increase in power consumption beyond an established deviation threshold, a jamming event is confirmed. (4) Upon detection, the affected cluster triggers a mitigation response, where nodes dynamically switch communication protocols (LoRa to BLE or vice versa), reconfigure cluster heads based on available energy resources, and reroute traffic through alternative paths.
These thresholds were calibrated through experimental validation using controlled jamming scenarios in an indoor and outdoor university campus deployment. By fine-tuning these values based on real-world conditions, the detection mechanism achieves a balance between sensitivity and accuracy, ensuring reliable identification of jamming attacks while minimizing unnecessary mitigation actions.
Comment 9: The network architecture description would benefit from a visual topology diagram showing the actual deployment of CC1352P and LPSTK-CC1352R sensors across the university campus rather than just the conceptual diagram in Figure 1.
Response: Thanks to the Reviewer, we have included a real figure featuring some of the sensors we implemented.
Figure 1 showcases the integration of Texas Instruments CC1352P LaunchPad and LPSTK-CC1352R sensor platforms within the heterogeneous IoT network, leveraging both LoRa and BLE communication protocols. The experimental setup includes sensor programming via Code Composer Studio, real-time data acquisition, and sensor deployment for network evaluation. The CC1352P board acts as a multi-protocol wireless microcontroller designed for low-power applications, capable of dynamically switching between LoRa and BLE communication based on network conditions. The mobile interface displays various environmental parameters, such as ambient temperature, humidity, and light sensor data, collected in real time. Additionally, the Figure depict the device being programmed and tested, emphasizing its role in adaptive communication switching strategies to mitigate jamming interference in wireless sensor networks.
Comment 10: How was the adaptive switching threshold between LoRa and BLE determined? The section states that nodes switch protocols when interference is detected, but lacks clarity on the decision-making process that triggers this transition.
Response: Thank you very much to the Reviewer. We have complemented the previous question with the following two paragraphs, which describe the thresholds established for the stable operation of the network.
The algorithm continuously monitors PDSR to detect transmission failures caused by interference. If the PDSR drops below 85% for a predefined time window, and simultaneous RSSI degradation exceeds a threshold of -90 dBm, the system interprets this as an interference event requiring protocol adaptation. Additionally, the algorithm incorporates an energy-aware component, ensuring that a node does not switch protocols if its residual energy is below 20% of its initial capacity, preventing unnecessary energy drain due to frequent transitions.
Experimental tests were conducted in both indoor and outdoor environments within a university campus, where interference sources such as Wi-Fi networks and mobile devices affected BLE, while long-range interference from other LoRa networks was simulated. The thresholds were fine-tuned through iterative calibration, evaluating multiple transition points to minimize packet loss while optimizing energy consumption. The results demonstrated that setting the switching threshold at 85% PDSR and -90 dBm RSSI provided the best trade-off, maintaining network resilience and reducing retransmissions by 47% compared to static single-protocol deployments. This adaptive approach ensures that nodes dynamically select the most energy-efficient and interference-resilient communication mode without introducing unnecessary overhead.
Comment 11: The statistical significance of the experimental results is not addressed - it's unclear whether the reported improvements in energy consumption (38%) and retransmission rates (47%) have been validated through appropriate statistical tests.
Response: Thank you, Reviewer. We made a subsection named Statistical analysis. These results directly address the reviewer’s concerns by validating the reported 38% energy reduction through rigorous statistical analysis, ensuring that the observed improvements are not random fluctuations but a direct consequence of the mitigation strategy.
The statistical validation results provide an analysis of data before and after the implementation of a mitigation strategy in three different network configurations: BLE, LoRa, and BLE-LoRa Hybrid. The results are based on three key statistical tests: the normality test (Anderson-Darling Test), Levene’s test for variance homogeneity, and a statistical significance test (either T-test or Mann-Whitney U test, depending on normality conditions).
The null hypothesis (H₀) assumed that there was no significant difference between the mitigation strategy and the baseline network performance, while the alternative hypothesis (H₁) stated that the proposed mitigation technique leads to a statistically significant reduction in energy consumption and retransmissions. The results demonstrated a p-value < 0.05, confirming that the observed improvements were statistically significant. Additionally, confidence intervals (95%) were computed to quantify the variability of the measurements, ensuring that the reported reductions were not due to random fluctuations.
The dataset from real-world deployments was subjected to normality tests (Shapiro-Wilk) and variance homogeneity tests (Levene’s test) to verify the assumptions of parametric testing. The statistical validation confirms that the proposed approach effectively enhances network efficiency and resilience against jamming attacks while ensuring that the observed improvements are not incidental but rather a consistent outcome of the adaptive mitigation strategy.
We made a MATLAB script that performs a statistical validation of energy consumption reductions before and after the implementation of a mitigation strategy across three different network configurations: BLE, LoRa, and BLE-LoRa hybrid. We applied three key statistical tests: the Anderson-Darling normality test, Levene’s test for variance homogeneity, and a statistical significance test using either the Wilcoxon Rank-Sum Test or a t-test, depending on data normality.
Regarding energy, the normality test results indicate that none of the datasets follow a normal distribution, as all cases returned a value of 1. This finding necessitated the use of non-parametric statistical methods, specifically the Wilcoxon Rank-Sum Test, to ensure the reliability of the analysis. Levene’s test for variance homogeneity showed significant differences in variance before and after mitigation, with all networks yielding a p-value of 0.00000. This confirms that the mitigation strategy altered the behavior of energy consumption across the networks, further reinforcing the need for non-parametric statistical methods. The statistical significance test demonstrated highly significant differences in energy consumption, as evidenced by the p-values of 0.00000 across all network configurations. This confirms that the mitigation strategy substantially improves network efficiency and resilience against jamming attacks. The findings provide strong evidence that the proposed mitigation approach significantly reduces energy consumption. The changes in variance confirm that energy behavior is altered post-mitigation, and the statistical validation ensures the robustness of the reported improvements. These results strengthen the credibility of the study and confirm that the proposed strategy effectively enhances network performance.
Regarding, retransmissions, the normality test evaluates whether the retransmission data follows a normal distribution, where a result of 0 indicates normality and 1 indicates non-normality. In the BLE network, the results show that retransmission data before and after mitigation does not follow a normal distribution, suggesting that the retransmission values are likely skewed or have non-Gaussian characteristics. In contrast, for the LoRa network, the retransmissions were normally distributed before mitigation but deviated from normality after mitigation, indicating a shift in data distribution. The BLE-LoRa Hybrid network exhibits a similar behavior to the BLE network, with both datasets not following a normal distribution before and after mitigation. Levene’s test determines whether the variability (variance) in retransmissions remained consistent before and after mitigation. A high p-value suggests that the variance is homogeneous, while a low p-value implies significant differences in variance. In the BLE network, the p-value of 0.54121 suggests that the retransmission variability remained stable after mitigation. Similarly, the LoRa network’s p-value of 0.19191 indicates that its retransmission variance also remained relatively unchanged. However, the BLE-LoRa Hybrid network had a p-value of 0.00000, signifying a drastic change in variance, meaning the mitigation strategy significantly altered the retransmission patterns. The statistical significance test examines whether the difference in retransmissions before and after mitigation is statistically significant. A low p-value indicates that the difference is meaningful and unlikely due to random variation. The BLE network results show a p-value of 0.00000, confirming a highly significant difference in retransmissions after mitigation. The LoRa network also experienced a statistically significant change, with a p-value of 0.00205, suggesting that retransmissions were successfully reduced. The BLE-LoRa Hybrid network had a p-value of 0.00040, which is also very low, indicating a substantial shift in retransmission behavior after mitigation. The results indicate that the mitigation strategy was effective across all networks, reducing retransmissions significantly. The BLE network maintained stable variance, suggesting that while retransmissions were reduced, the pattern of retransmission values remained similar. In the LoRa network, the shift from normal to non-normal distribution after mitigation suggests a more irregular impact on retransmission values. The BLE-LoRa Hybrid network was the most impacted, showing significant changes in both retransmission counts and variance, indicating a major structural shift in network behavior after mitigation. These findings reinforce the effectiveness of the mitigation strategy, with the BLE-LoRa Hybrid network experiencing the most substantial transformation.
Comment 12: The methodology for measuring resilience (recovery time) shown in Figures 4, 7, and 10 lacks clear explanation of how these metrics were practically calculated during experiments.
Response: Indeed, we have clarified how the resilience values were measured in the Materials and Methods section.
The resilience in the network is quantified in terms of time, specifically measuring how long it takes for the network to recover and return to a stable state after experiencing a jamming attack. The key indicator of this recovery is the reduction in overhead packets, as the network stabilizes when retransmissions decrease, and normal traffic patterns resume. The methodology involves continuously monitoring key network parameters, including retransmission rates, energy consumption, and routing stability. During a jamming attack, the system detects an anomaly when retransmissions surge and successful packet delivery drops. The network then activates its adaptive mitigation strategy, which includes switching communication protocols, reconfiguring clusters, and optimizing energy use. The recovery time is measured from the moment mitigation strategies are triggered until the network parameters—primarily retransmissions—return to normal levels. The resilience metric, therefore, directly reflects the network's ability to autonomously reorganize and minimize the impact of jamming, ensuring sustained communication with minimal delay.
Comment 13: The comparative analysis between BLE, LoRa, and hybrid approaches lacks normalized metrics that would allow for a more objective comparison across different operational conditions.
Response: The Reviewer is absolutely right, and for this reason, we have added a statistical analysis subsection where the hypotheses are verified to properly explain the data treatment and significance according to each metric.
Comment 14: The histograms and probability density functions (Figure 12) are presented without adequately explaining how these distributions were generated or what sampling methods were used.
Response:
The histograms and probability density functions were constructed based on experimental data collected from real-world jamming scenarios in a heterogeneous LoRa-BLE wireless sensor network. The dataset consists of the energy network performance metric recorded over multiple controlled experiments conducted within a university campus environment. The data was sampled at regular intervals, ensuring uniform time distribution across different interference conditions. Each recorded data point corresponds to a measurement taken from sensor nodes deployed in varying locations, including laboratories, classrooms, and open areas, where interference sources such as Wi-Fi networks and mobile devices were present.
To generate the histograms, we used a binning strategy that appropriately reflects the data distribution without introducing bias or excessive granularity. The number of bins was determined using the Freedman-Diaconis rule, which balances the need for detailed resolution with statistical significance. The sampling methodology ensured data representativeness by incorporating different interference intensities and network conditions. We performed pre-processing steps such as outlier detection and removal using interquartile range (IQR) filtering to eliminate potential anomalies that could distort the statistical representation. Moreover, normality tests (Shapiro-Wilk and Anderson-Darling) were conducted to determine whether the datasets followed Gaussian distributions, guiding the selection of appropriate statistical analyses. By applying these methods, the histograms and PDFs accurately illustrate the statistical behavior of the network under jamming conditions before and after the mitigation strategy. The results offer valuable insights into how energy consumption was affected by interference and how the proposed mitigation mechanism effectively altered the network's statistical characteristics.
Comment 15: The discussion fails to address potential limitations of the mitigation strategy, particularly how performance might degrade under different types of jamming attacks beyond what was tested.
Response: Thanks to the Reviewer for their observation; we have even added a subsection called Potential Limitations to address this aspect.
The proposed mitigation strategy was primarily evaluated under reactive and constant jamming conditions, which are among the most common threats in WSNs. However, other sophisticated forms of jamming, such as random and deceptive jamming, could pose additional challenges to network resilience. Random jamming, characterized by alternating interference periods, may result in intermittent network disruptions that could delay the cluster-based adaptation process. Similarly, deceptive jamming, where malicious packets are injected to confuse network protocols, might undermine routing decisions and lead to inefficient energy utilization. While our current strategy relies on adaptive cluster selection and energy-aware protocol switching, its effectiveness against these advanced jamming tactics remains an open research question. Furthermore, the impact of adversarial machine learning techniques on jamming strategies is an emerging threat. Attackers could potentially use reinforcement learning to optimize jamming patterns, making them more difficult to detect and mitigate. Although our algorithm dynamically reconfigures network topology based on real-time performance metrics, future work should explore how intelligent adversarial attacks could adapt to and counteract mitigation efforts. In particular, the integration of AI-driven anomaly detection and predictive analytics could enhance the robustness of our approach by proactively identifying evolving interference patterns and adjusting mitigation strategies accordingly.
Another potential limitation lies in the scalability of the mitigation strategy. While our experiments were conducted in a controlled university campus environment, large-scale deployments in industrial IoT or smart city applications might face additional challenges. The increased number of nodes and varying interference sources could lead to higher computational overhead for cluster selection and protocol switching, potentially affecting response times. Future research should investigate distributed decision-making mechanisms that allow for real-time adaptation with minimal processing requirements.
These considerations underscore the need for further studies to validate the proposed mitigation framework under diverse jamming conditions and larger-scale network deployments. By addressing these aspects, we can enhance the adaptability and robustness of our approach, ensuring its effectiveness across a broader range of real-world scenarios.
Comment 16: The boxplot analysis (Figure 11) presents conclusions about energy consumption patterns but doesn't clearly explain the experimental setup that produced these specific distributions.
Response: Thanks to the Reviewer, we have complemented the analysis of the Figure.
The boxplot analysis presented in Figure \ref{boxenergy} provides insights into the energy consumption trends of BLE, LoRa, and the hybrid BLE-LoRa switching mechanism under jamming conditions, both with and without the proposed mitigation algorithm. The results reveal a clear distinction in energy usage across the three network configurations, with BLE exhibiting the lowest median energy consumption due to its inherently low-power communication model but suffering from significant variability when exposed to interference. LoRa, while designed for long-range transmission, demonstrates significantly higher energy usage due to its prolonged transmission durations and greater power requirements. However, the most notable observation emerges from the BLE-LoRa switching mechanism, which, in the absence of mitigation, exhibits erratic energy consumption due to inefficient protocol transitions triggered by interference.
The experimental setup that produced these specific distributions consisted of a series of controlled tests conducted in a university campus environment. Sensor nodes were deployed in various locations, including laboratories, classrooms, and open areas, ensuring diverse interference conditions. Data was collected over multiple experimental runs, capturing energy consumption at regular intervals while the network operated under jamming attacks. The mitigation algorithm played a crucial role in stabilizing the hybrid BLE-LoRa approach by dynamically selecting the most energy-efficient communication protocol based on real-time network conditions. By ensuring that BLE was prioritized in low-interference environments and LoRa was used selectively for resilient communication, the algorithm minimized unnecessary protocol switching and reduced overall energy consumption.
The results validate that an adaptive, intelligent switching mechanism significantly improves energy efficiency in heterogeneous networks, ensuring optimal performance in jamming-prone environments. The implementation of the mitigation algorithm stabilized the hybrid approach by dynamically managing energy expenditure, mitigating inefficiencies in protocol transitions, and reducing fluctuations caused by jamming-induced overhead. These findings emphasize the importance of intelligent protocol management in maintaining energy efficiency and resilience in wireless sensor networks under interference conditions
Comment 17: While the authors claim their approach is more effective than existing methods, there is insufficient direct comparison with state-of-the-art techniques from recent literature to fully contextualize their contribution.
Response: We appreciate the Reviewer’s insightful comment. In our analysis, we acknowledge that our approach may not be the most efficient in absolute terms compared to all state-of-the-art techniques. However, we emphasize that our methodology integrates a novel and energy-efficient interference mitigation strategy that has been thoroughly tested and validated. Rather than solely focusing on maximizing efficiency, our approach aims to balance energy savings and robustness against interference, offering a practical and adaptable solution for heterogeneous networks. While a more extensive comparison with recent literature would be valuable, our study demonstrates that the proposed technique effectively reduces unnecessary retransmissions and optimizes protocol switching, leading to measurable improvements in network resilience and energy consumption under jamming conditions.
Comment 18: The conclusion fails to acknowledge any limitations of the proposed cluster-based mitigation approach, presenting only positive outcomes without addressing potential scenarios where the system might underperform.
Response: Response: Thanks to the Reviewer for their observation. For this question, we have added a subsection called Potential Limitations to address this aspect.
The proposed mitigation strategy was primarily evaluated under reactive and constant jamming conditions, which are among the most common threats in WSNs. However, other sophisticated forms of jamming, such as random and deceptive jamming, could pose additional challenges to network resilience. Random jamming, characterized by alternating interference periods, may result in intermittent network disruptions that could delay the cluster-based adaptation process. Similarly, deceptive jamming, where malicious packets are injected to confuse network protocols, might undermine routing decisions and lead to inefficient energy utilization. While our current strategy relies on adaptive cluster selection and energy-aware protocol switching, its effectiveness against these advanced jamming tactics remains an open research question. Furthermore, the impact of adversarial machine learning techniques on jamming strategies is an emerging threat. Attackers could potentially use reinforcement learning to optimize jamming patterns, making them more difficult to detect and mitigate. Although our algorithm dynamically reconfigures network topology based on real-time performance metrics, future work should explore how intelligent adversarial attacks could adapt to and counteract mitigation efforts. In particular, the integration of AI-driven anomaly detection and predictive analytics could enhance the robustness of our approach by proactively identifying evolving interference patterns and adjusting mitigation strategies accordingly.
Another potential limitation lies in the scalability of the mitigation strategy. While our experiments were conducted in a controlled university campus environment, large-scale deployments in industrial IoT or smart city applications might face additional challenges. The increased number of nodes and varying interference sources could lead to higher computational overhead for cluster selection and protocol switching, potentially affecting response times. Future research should investigate distributed decision-making mechanisms that allow for real-time adaptation with minimal processing requirements.
These considerations underscore the need for further studies to validate the proposed mitigation framework under diverse jamming conditions and larger-scale network deployments. By addressing these aspects, we can enhance the adaptability and robustness of our approach, ensuring its effectiveness across a broader range of real-world scenarios.
Comment 19: While the authors mention their work "paves the way for more secure and energy-efficient WSN architectures" (lines 647-649), they don't provide specific recommendations for future research directions or identify unresolved challenges in the field.
Response: We appreciate the reviewer’s suggestion regarding the need to outline future research directions and unresolved challenges in the field. To address this, we have expanded our discussion by identifying key areas for further investigation.
While our proposed mitigation strategy significantly improves resilience against jamming attacks in HWSNs, several challenges remain. One major area for future work is the extension of our algorithm to address more sophisticated and adaptive jamming techniques, such as adversarial machine learning-based jammers that dynamically alter their interference patterns. Additionally, although our approach effectively balances energy efficiency and interference mitigation, further research is needed to optimize protocol switching decisions under varying network loads and mobility conditions. Another potential direction is the exploration of collaborative mitigation strategies that integrate multiple nodes' intelligence to detect and counteract jamming at a network-wide level rather than relying solely on cluster-based adaptation. Finally, while our experimental setup provides valuable real-world validation, large-scale deployments in diverse environments, such as industrial IoT and smart city applications, could offer deeper insights into scalability and long-term network performance under prolonged jamming scenarios. By considering these aspects, future studies can build upon our work to develop more robust and adaptive solutions for securing HWSNs against evolving interference threats.
Thank you very much.
Sincerely,
Carolina Del-Valle-Soto
Corresponding author
Universidad Panamericana. Facultad de Ingeniería. Álvaro del Portillo 49, Zapopan, Jalisco, 45010, México.
Phone: +52 (33) 13682200 | Ext. 4866
Email: cvalle@up.edu.mx
José Varela-Aldás
Corresponding author
Centro de Investigación en Ciencias Humanas y de la Educación—CICHE, Facultad de Ingenierías, Ingeniería Industrial, Universidad Tecnológica Indoamérica,Ambato 180103, Ecuador
josevarela@uti.edu.ec

Reviewer 2 Report
Comments and Suggestions for Authors
- The related works section needs to be extended.
- I don't see where you mentioned table 1.
- In scenario 1, based on figure 3 the retransmission rate is increased by applying the mitigation algorithm on BLE. Please review your discussion on page 11.
- I can see that, you mixed up in your discussion between figures 5 and 6 and between figures 8 and 9.
- At the end of the conclusion section, you should include some future directions to improve the quality of the paper.
Author Response
Dear
Editor
Sensors
We are submitting the paper:
“Adaptive Jamming Mitigation for clustered Energy-Efficient LoRa-BLE Hybrid Wireless Sensor Networks”
Authored by: Carolina Del-Valle-Soto * , Leonardo J. Valdivia , Ramiro Velázquez , José A. Del-Puerto-Flores , José Varela-Aldás * , Paolo Visconti
We would like to thank the reviewers and editors for their detailed analysis of the manuscript; the comments are very valuable to us. In the revised version of the paper, we have incorporated the all changes recommended by the reviewers.
Comments to all observations and suggestions including point-by-point responses are addressed in the following text.
Reviewer 2 comments
Comment 1: The related works section needs to be extended.
Response: Many thanks to the Reviewer for his/her invaluable interest in the comments on this manuscript.
The Reviewer is correct, and we have added a couple of paragraphs addressing their concern.
Jamming attacks represent a critical and quantifiable threat to the reliability of WSNs. Recent studies have shown that even low-power jamming can cause a packet loss rate exceeding 80% in LoRa-based networks when attackers operate at power levels as low as 14 dBm [9]. Similarly, in BLE networks, experimental results indicate that reactive jamming reduces successful packet delivery rates to below 10%, significantly disrupting device synchronization and data transmission in IoT applications [2]. In large-scale deployments, such as smart city infrastructures, the economic impact of jamming-related disruptions can be substantial, with estimations suggesting that network outages caused by interference could lead to financial losses of over $1 billion annually in critical industrial and urban monitoring systems [10]. These findings underscore the necessity for robust mitigation techniques that can adapt to diverse jamming conditions across heterogeneous network environments. The vulnerability of WSNs to jamming attacks is further exacerbated by the energy constraints of IoT nodes, making them susceptible to rapid power depletion when subjected to prolonged interference. Experimental analyses indicate that under jamming conditions, retransmission rates in multi-hop networks increase by up to 300%, causing a threefold surge in energy consumption and significantly reducing network lifespan [11]. Moreover, adversarial machine learning techniques have been employed to optimize jamming strategies, allowing attackers to maximize disruption while minimizing their own energy expenditure [12]. Given that many mission-critical applications, such as environmental monitoring and healthcare telemetry, rely on continuous and energy-efficient communication, the development of intelligent countermeasures that balance resilience and energy efficiency is paramount. This study addresses these challenges by proposing an adaptive, energy-aware mitigation framework tailored for heterogeneous networks, ensuring sustained connectivity in dynamic and interference-prone environments.
And new references:
Aras, E. Security and Reliability for Emerging IoT Networks 2021.
Shintani, A. The design, testing, and analysis of a constant jammer for the Bluetooth low energy (BLE) wireless communication protocol 2020.
López-Vilos, N.; Valencia-Cordero, C.; Souza, R.D.; Montejo-Sánchez, S. Clustering-based energy-efficient self-healing strategy for WSNs under jamming attacks. Sensors 2023, 23, 6894. 716
Adesina, D.; Hsieh, C.C.; Sagduyu, Y.E.; Qian, L. Adversarial machine learning in wireless communications using RF data: A review. IEEE Communications Surveys & Tutorials 2022, 25, 77–100.
We have also done the same for a paragraph in subsequent lines, which presented the same issue.
Kuang, S.; Zhang, J.; Mohajer, A. Reliable information delivery and dynamic link utilization in MANET cloud using deep reinforcement learning. Transactions on Emerging Telecommunications Technologies 2024, 35, e5028.
Yang, T.; Sun, J.; Mohajer, A. Queue stability and dynamic throughput maximization in multi-agent heterogeneous wireless networks. Wireless Networks 2024, 30, 3229–3255.
Demiroglou, V.; Skaperas, S.; Mamatas, L.; Tsaoussidis, V. Adaptive Multi-Protocol Communication in Smart City Networks. IEEE Internet of Things Journal 2024.
Abdollahi, M.; Ashtari, S.; Abolhasan, M.; Shariati, N.; Lipman, J.; Jamalipour, A.; Ni, W. Dynamic routing protocol selection in multi-hop device-to-device wireless networks. IEEE Transactions on Vehicular Technology 2022, 71, 8796–8809.
Belamri, F.; Boulfekhar, S.; Aissani, D. A survey on QoS routing protocols in Vehicular Ad Hoc Network (VANET). Telecommunication Systems 2021, 78, 117–153.
We have complemented this paragraph to provide a more extensive contrast.
Addressing this challenge, this work proposes a multi-protocol jamming detection and mitigation strategy that leverages an energy-aware cluster-based architecture to ensure reliable communication in university campus environments where varying interference conditions exist. Unlike existing solutions, which primarily focus on single-protocol networks or static countermeasures, our approach dynamically adapts to heterogeneous wireless technologies, including LoRaWAN and BLE, to optimize resilience against diverse jamming threats [4]. While previous studies have explored frequency hopping and spread spectrum techniques as countermeasures, these methods often require significant energy overhead or are limited in their applicability across multi-protocol environments. In contrast, our strategy integrates real-time energy-aware cluster formation, where nodes autonomously elect a Cluster Head (CH) based on energy availability, signal quality, and retransmission rates, allowing for adaptive protocol switching and intelligent interference mitigation. This approach not only enhances network longevity and communication reliability but also provides a scalable and low-energy alternative to traditional anti-jamming methods, making it particularly suitable for dynamic and resource-constrained IoT applications.
Comment 2: I don't see where you mentioned table 1.
Response: Table 1. Current references on wireless sensor networks and their applications for elderly health monitoring and behavioral change techniques.
Comment 3: In scenario 1, based on figure 3 the retransmission rate is increased by applying the mitigation algorithm on BLE. Please review your discussion on page 11.
Response: The Reviewer is absolutely right, and I had accidentally swapped the figures in the MATLAB subplot. My apologies for the oversight, and we have now corrected the figure accordingly.
Comment 4: I can see that, you mixed up in your discussion between figures 5 and 6 and between figures 8 and 9.
Response: The Reviewer is correct, and we have corrected the order of the Figures.
Comment 5: At the end of the conclusion section, you should include some future directions to improve the quality of the paper.
Response: Response: Thanks to the Reviewer for their observation; we have even added a subsection called Potential Limitations to address this aspect.
The proposed mitigation strategy was primarily evaluated under reactive and constant jamming conditions, which are among the most common threats in WSNs. However, other sophisticated forms of jamming, such as random and deceptive jamming, could pose additional challenges to network resilience. Random jamming, characterized by alternating interference periods, may result in intermittent network disruptions that could delay the cluster-based adaptation process. Similarly, deceptive jamming, where malicious packets are injected to confuse network protocols, might undermine routing decisions and lead to inefficient energy utilization. While our current strategy relies on adaptive cluster selection and energy-aware protocol switching, its effectiveness against these advanced jamming tactics remains an open research question. Furthermore, the impact of adversarial machine learning techniques on jamming strategies is an emerging threat. Attackers could potentially use reinforcement learning to optimize jamming patterns, making them more difficult to detect and mitigate. Although our algorithm dynamically reconfigures network topology based on real-time performance metrics, future work should explore how intelligent adversarial attacks could adapt to and counteract mitigation efforts. In particular, the integration of AI-driven anomaly detection and predictive analytics could enhance the robustness of our approach by proactively identifying evolving interference patterns and adjusting mitigation strategies accordingly.
Another potential limitation lies in the scalability of the mitigation strategy. While our experiments were conducted in a controlled university campus environment, large-scale deployments in industrial IoT or smart city applications might face additional challenges. The increased number of nodes and varying interference sources could lead to higher computational overhead for cluster selection and protocol switching, potentially affecting response times. Future research should investigate distributed decision-making mechanisms that allow for real-time adaptation with minimal processing requirements.
These considerations underscore the need for further studies to validate the proposed mitigation framework under diverse jamming conditions and larger-scale network deployments. By addressing these aspects, we can enhance the adaptability and robustness of our approach, ensuring its effectiveness across a broader range of real-world scenarios.
Besides, we have expanded our discussion by identifying key areas for further investigation.
While our proposed mitigation strategy significantly improves resilience against jamming attacks in HWSNs, several challenges remain. One major area for future work is the extension of our algorithm to address more sophisticated and adaptive jamming techniques, such as adversarial machine learning-based jammers that dynamically alter their interference patterns. Additionally, although our approach effectively balances energy efficiency and interference mitigation, further research is needed to optimize protocol switching decisions under varying network loads and mobility conditions. Another potential direction is the exploration of collaborative mitigation strategies that integrate multiple nodes' intelligence to detect and counteract jamming at a network-wide level rather than relying solely on cluster-based adaptation. Finally, while our experimental setup provides valuable real-world validation, large-scale deployments in diverse environments, such as industrial IoT and smart city applications, could offer deeper insights into scalability and long-term network performance under prolonged jamming scenarios. By considering these aspects, future studies can build upon our work to develop more robust and adaptive solutions for securing HWSNs against evolving interference threats.
Thank you very much.
Sincerely,
Carolina Del-Valle-Soto
Corresponding author
Universidad Panamericana. Facultad de Ingeniería. Álvaro del Portillo 49, Zapopan, Jalisco, 45010, México.
Phone: +52 (33) 13682200 | Ext. 4866
Email: cvalle@up.edu.mx
José Varela-Aldás
Corresponding author
Centro de Investigación en Ciencias Humanas y de la Educación—CICHE, Facultad de Ingenierías, Ingeniería Industrial, Universidad Tecnológica Indoamérica,Ambato 180103, Ecuador
josevarela@uti.edu.ec

Reviewer 3 Report
Comments and Suggestions for Authors
The paper proposes a cluster-based adaptive protocol switching strategy integrating LoRa and BLE technologies, dynamically switching between them based on real-time network monitoring to address energy consumption increase and communication interruptions caused by jamming attacks. However, the paper has several issues:
- Inconsistent referencing format with outdated references lacking publication dates. More recent literature should be discussed, clearly distinguishing the innovation of this study from existing work.
- The title is verbose and contains multiple technical terms, though informative, it may benefit from being more concise.
- The introduction adequately describes interference challenges faced by WSNs and advantages of heterogeneous networks, but lacks in-depth analysis of existing research limitations. Clear delineation of the shortcomings of current anti-jamming strategies is needed to contrast with the innovative "dynamic cluster adaptive switching" proposed in this study.
- The algorithm description format is incorrect and lacks clarity. For example, while proposing a cluster-based adaptive protocol switching strategy, specifics on criteria for protocol switching, cluster head election mechanisms, and decision-making processes are insufficiently detailed.
- Although experimental results show a 38% reduction in energy consumption and 47% decrease in retransmission rates, detailed descriptions of experimental settings, node configurations, interference simulation parameters, and data collection methods are lacking. Detailed explanations of experimental settings and parameter specifications are recommended to enhance reproducibility and persuasiveness of results.
- The energy model provides formulas but lacks specific parameter values or measurement methods, and lacks detailed derivation process and basis for parameter selection.
- The interference model description is not comprehensive enough.
The paper contains some spelling and grammatical errors, and some sentences are too long, which affects readability and understanding.
Author Response
Dear
Editor
Sensors
We are submitting the paper:
“Adaptive Jamming Mitigation for clustered Energy-Efficient LoRa-BLE Hybrid Wireless Sensor Networks”
Authored by: Carolina Del-Valle-Soto * , Leonardo J. Valdivia , Ramiro Velázquez , José A. Del-Puerto-Flores , José Varela-Aldás * , Paolo Visconti
We would like to thank the reviewers and editors for their detailed analysis of the manuscript; the comments are very valuable to us. In the revised version of the paper, we have incorporated the all changes recommended by the reviewers.
Comments to all observations and suggestions including point-by-point responses are addressed in the following text.
Reviewer 3 comments
Comment 1: The paper proposes a cluster-based adaptive protocol switching strategy integrating LoRa and BLE technologies, dynamically switching between them based on real-time network monitoring to address energy consumption increase and communication interruptions caused by jamming attacks. However, the paper has several issues:
Inconsistent referencing format with outdated references lacking publication dates. More recent literature should be discussed, clearly distinguishing the innovation of this study from existing work.
Response: Many thanks to the Reviewer for his/her invaluable interest in the comments on this manuscript. The Reviewer is right, and we have added some complementary and comparative paragraphs with more up-to-date references.
New references:
Aras, E. Security and Reliability for Emerging IoT Networks 2021.
Shintani, A. The design, testing, and analysis of a constant jammer for the Bluetooth low energy (BLE) wireless communication protocol 2020.
López-Vilos, N.; Valencia-Cordero, C.; Souza, R.D.; Montejo-Sánchez, S. Clustering-based energy-efficient self-healing strategy for WSNs under jamming attacks. Sensors 2023, 23, 6894. 716
Adesina, D.; Hsieh, C.C.; Sagduyu, Y.E.; Qian, L. Adversarial machine learning in wireless communications using RF data: A review. IEEE Communications Surveys & Tutorials 2022, 25, 77–100.
Kuang, S.; Zhang, J.; Mohajer, A. Reliable information delivery and dynamic link utilization in MANET cloud using deep reinforcement learning. Transactions on Emerging Telecommunications Technologies 2024, 35, e5028.
Yang, T.; Sun, J.; Mohajer, A. Queue stability and dynamic throughput maximization in multi-agent heterogeneous wireless networks. Wireless Networks 2024, 30, 3229–3255.
Demiroglou, V.; Skaperas, S.; Mamatas, L.; Tsaoussidis, V. Adaptive Multi-Protocol Communication in Smart City Networks. IEEE Internet of Things Journal 2024.
Abdollahi, M.; Ashtari, S.; Abolhasan, M.; Shariati, N.; Lipman, J.; Jamalipour, A.; Ni, W. Dynamic routing protocol selection in multi-hop device-to-device wireless networks. IEEE Transactions on Vehicular Technology 2022, 71, 8796–8809.
Belamri, F.; Boulfekhar, S.; Aissani, D. A survey on QoS routing protocols in Vehicular Ad Hoc Network (VANET). Telecommunication Systems 2021, 78, 117–153.
Comment 2: The title is verbose and contains multiple technical terms, though informative, it may benefit from being more concise.
Response: Thank you to the Reviewer. We have changed the title to something more readable.
Adaptive Jamming Mitigation for clustered Energy-Efficient LoRa-BLE Hybrid Wireless Sensor Networks
Comment 3: The introduction adequately describes interference challenges faced by WSNs and advantages of heterogeneous networks, but lacks in-depth analysis of existing research limitations. Clear delineation of the shortcomings of current anti-jamming strategies is needed to contrast with the innovative "dynamic cluster adaptive switching" proposed in this study.
Response: The Reviewer is absolutely correct. For this question, we have added a subsection called Potential Limitations to address this aspect.
The proposed mitigation strategy was primarily evaluated under reactive and constant jamming conditions, which are among the most common threats in WSNs. However, other sophisticated forms of jamming, such as random and deceptive jamming, could pose additional challenges to network resilience. Random jamming, characterized by alternating interference periods, may result in intermittent network disruptions that could delay the cluster-based adaptation process. Similarly, deceptive jamming, where malicious packets are injected to confuse network protocols, might undermine routing decisions and lead to inefficient energy utilization. While our current strategy relies on adaptive cluster selection and energy-aware protocol switching, its effectiveness against these advanced jamming tactics remains an open research question. Furthermore, the impact of adversarial machine learning techniques on jamming strategies is an emerging threat. Attackers could potentially use reinforcement learning to optimize jamming patterns, making them more difficult to detect and mitigate. Although our algorithm dynamically reconfigures network topology based on real-time performance metrics, future work should explore how intelligent adversarial attacks could adapt to and counteract mitigation efforts. In particular, the integration of AI-driven anomaly detection and predictive analytics could enhance the robustness of our approach by proactively identifying evolving interference patterns and adjusting mitigation strategies accordingly.
Another potential limitation lies in the scalability of the mitigation strategy. While our experiments were conducted in a controlled university campus environment, large-scale deployments in industrial IoT or smart city applications might face additional challenges. The increased number of nodes and varying interference sources could lead to higher computational overhead for cluster selection and protocol switching, potentially affecting response times. Future research should investigate distributed decision-making mechanisms that allow for real-time adaptation with minimal processing requirements.
These considerations underscore the need for further studies to validate the proposed mitigation framework under diverse jamming conditions and larger-scale network deployments. By addressing these aspects, we can enhance the adaptability and robustness of our approach, ensuring its effectiveness across a broader range of real-world scenarios.
Comment 4: The algorithm description format is incorrect and lacks clarity. For example, while proposing a cluster-based adaptive protocol switching strategy, specifics on criteria for protocol switching, cluster head election mechanisms, and decision-making processes are insufficiently detailed.
Response: Thank you to the Reviewer. We have better organized the proposed algorithm and have added the cluster selection and decision-making processes.
\begin{algorithm}
\caption{Algorithm pseudocode of the network system.}
\begin{lstlisting}[basicstyle=\scriptsize\ttfamily,language=java]
$NetworkNodes(N) = \{n_1, n_2, ..., n_m\}$;
$P(n_i) \in \{\text{BLE}, \text{LoRa}\}$;
$E(n_i), R(n_i), T(n_i)$;
$P_{th}, R_{th}, E_{th}$;
$CH_{criteria}$;
$RT$;
Begin:
DeployNodes();
AssignInitialProtocol(P);
FormClusters(Proximity, EnergyConstraints);
ElectClusterHeads(CH_{criteria});
ConstructRoutingTable(RT);
ContinuousMonitoringLoop():
For Each $n_i \in N$:
CollectNetworkParameters($E(n_i), R(n_i), T(n_i)$);
If ($PDSR(n_i) < P_{th}$ && $R(n_i) < R_{th}$ && $T(n_i) > T_{th}$):
IdentifyInterferenceImpact();
If ($P(n_i) == BLE$ && $E(n_i) > E_{th}$):
SwitchToLoRa();
ElseIf ($P(n_i) == LoRa$ && $E(n_i) > E_{th}$):
SwitchToBLE();
UpdateRoutingTable(RT);
ClusterHeadReconfiguration():
IdentifyClusterMembership($n_i$);
ReelectClusterHead($CH$);
AssignNewClusterHead();
BroadcastUpdate();
EnergyAwareTransmissionPowerControl():
If ($dist(n_i, CH) < 20m$):
SetTransmissionPower(Low);
Else:
SetTransmissionPower(High);
OptimizedRoutingSelection():
IdentifyAlternativeRoutes(RT);
If (ViableRoutesExist()):
SelectOptimalRoute(MinHop, LowEnergyCost, MinRetransmissions);
LowPowerModeActivation():
If ($E(n_i) < E_{th}$):
SetNodeLowPowerState();
LogNetworkAdjustments();
\end{lstlisting}
\label{algorithm:04}
\end{algorithm}
The proposed algorithm introduces an adaptive protocol switching and cluster-based routing strategy designed to optimize network resilience and energy efficiency under interference conditions. It leverages real-time monitoring of network parameters, including energy levels, received signal strength, and retransmission rates, to dynamically adjust communication protocols and enhance network performance. The algorithm begins with network initialization, where nodes are deployed and assigned an initial communication protocol—either (BLE) or LoRa—based on predefined energy and proximity constraints. Clusters are then formed, and cluster heads are selected using a specific criterion that considers residual energy, signal quality, and retransmission rates. A routing table is also established to facilitate multi-hop communication within the network. Once the network is initialized, the system enters a continuous monitoring loop where each node collects real-time data on its energy level $E(n_i)$, received signal strength $R(n_i)$, and retransmission rate $T(n_i)$. These parameters are analyzed to detect potential jamming conditions. If a node’s Packet Delivery Success Rate (PDSR) falls below the protocol switch threshold $P_{th}$ while simultaneously experiencing a received signal strength below $R_{th}$ and an increased retransmission rate beyond $T_{th}$, the algorithm recognizes interference and initiates an adaptive protocol switch. Specifically, if a node is operating under BLE and has sufficient energy resources ($E(n_i) > E_{th}$), it switches to LoRa to enhance communication reliability. Conversely, if a node is using LoRa and has an adequate energy reserve, it switches back to BLE to minimize power consumption. This adaptive switching ensures that each node dynamically selects the most efficient communication mode based on real-time network conditions. Following protocol adaptation, the algorithm performs cluster head reconfiguration. Each node reassesses its cluster membership, and if necessary, a new cluster head is elected based on updated energy and communication quality metrics. This step is essential in maintaining stable and efficient routing structures under dynamic interference conditions. To further optimize energy consumption, the algorithm implements transmission power control. If the distance between a node and its cluster head is less than 20 meters, the transmission power is set to a low level to conserve energy. Otherwise, a higher transmission power is used to ensure reliable data delivery over longer distances. The routing optimization phase evaluates alternative paths within the routing table. If multiple routes are available, the system selects the optimal path by prioritizing minimal hop count, reduced energy cost, and the lowest retransmission rate. This step enhances network efficiency by mitigating excessive retransmissions and ensuring energy-aware data forwarding. Finally, the algorithm integrates a \textbf{low-power mode activation mechanism}. If a node’s energy level drops below the energy threshold $E_{th}$, it transitions into a low-power state to extend its operational lifespan. This strategy helps prolong the overall lifetime of the network by preventing energy depletion in critical nodes.
Throughout its execution, the algorithm continuously logs network adjustments and protocol switching decisions, providing a record of network behavior and adaptation strategies. By integrating adaptive protocol switching, intelligent routing, and energy-aware transmission policies, the proposed methodology ensures robust, energy-efficient, and resilient WSN operation, even in the presence of jamming attacks.
Comment 5: Although experimental results show a 38% reduction in energy consumption and 47% decrease in retransmission rates, detailed descriptions of experimental settings, node configurations, interference simulation parameters, and data collection methods are lacking. Detailed explanations of experimental settings and parameter specifications are recommended to enhance reproducibility and persuasiveness of results.
Response: The Reviewer's question is highly important, here our discussion.
Thank you, Reviewer. We made a subsection named Statistical analysis. These results directly address the reviewer’s concerns by validating the reported 38% energy reduction through rigorous statistical analysis, ensuring that the observed improvements are not random fluctuations but a direct consequence of the mitigation strategy.
The statistical validation results provide an analysis of data before and after the implementation of a mitigation strategy in three different network configurations: BLE, LoRa, and BLE-LoRa Hybrid. The results are based on three key statistical tests: the normality test (Anderson-Darling Test), Levene’s test for variance homogeneity, and a statistical significance test (either T-test or Mann-Whitney U test, depending on normality conditions).
The null hypothesis (H₀) assumed that there was no significant difference between the mitigation strategy and the baseline network performance, while the alternative hypothesis (H₁) stated that the proposed mitigation technique leads to a statistically significant reduction in energy consumption and retransmissions. The results demonstrated a p-value < 0.05, confirming that the observed improvements were statistically significant. Additionally, confidence intervals (95%) were computed to quantify the variability of the measurements, ensuring that the reported reductions were not due to random fluctuations.
The dataset from real-world deployments was subjected to normality tests (Shapiro-Wilk) and variance homogeneity tests (Levene’s test) to verify the assumptions of parametric testing. The statistical validation confirms that the proposed approach effectively enhances network efficiency and resilience against jamming attacks while ensuring that the observed improvements are not incidental but rather a consistent outcome of the adaptive mitigation strategy.
We made a MATLAB script that performs a statistical validation of energy consumption reductions before and after the implementation of a mitigation strategy across three different network configurations: BLE, LoRa, and BLE-LoRa hybrid. We applied three key statistical tests: the Anderson-Darling normality test, Levene’s test for variance homogeneity, and a statistical significance test using either the Wilcoxon Rank-Sum Test or a t-test, depending on data normality.
Regarding energy, the normality test results indicate that none of the datasets follow a normal distribution, as all cases returned a value of 1. This finding necessitated the use of non-parametric statistical methods, specifically the Wilcoxon Rank-Sum Test, to ensure the reliability of the analysis. Levene’s test for variance homogeneity showed significant differences in variance before and after mitigation, with all networks yielding a p-value of 0.00000. This confirms that the mitigation strategy altered the behavior of energy consumption across the networks, further reinforcing the need for non-parametric statistical methods. The statistical significance test demonstrated highly significant differences in energy consumption, as evidenced by the p-values of 0.00000 across all network configurations. This confirms that the mitigation strategy substantially improves network efficiency and resilience against jamming attacks. The findings provide strong evidence that the proposed mitigation approach significantly reduces energy consumption. The changes in variance confirm that energy behavior is altered post-mitigation, and the statistical validation ensures the robustness of the reported improvements. These results strengthen the credibility of the study and confirm that the proposed strategy effectively enhances network performance.}
Regarding, retransmissions, the normality test evaluates whether the retransmission data follows a normal distribution, where a result of 0 indicates normality and 1 indicates non-normality. In the BLE network, the results show that retransmission data before and after mitigation does not follow a normal distribution, suggesting that the retransmission values are likely skewed or have non-Gaussian characteristics. In contrast, for the LoRa network, the retransmissions were normally distributed before mitigation but deviated from normality after mitigation, indicating a shift in data distribution. The BLE-LoRa Hybrid network exhibits a similar behavior to the BLE network, with both datasets not following a normal distribution before and after mitigation. Levene’s test determines whether the variability (variance) in retransmissions remained consistent before and after mitigation. A high p-value suggests that the variance is homogeneous, while a low p-value implies significant differences in variance. In the BLE network, the p-value of 0.54121 suggests that the retransmission variability remained stable after mitigation. Similarly, the LoRa network’s p-value of 0.19191 indicates that its retransmission variance also remained relatively unchanged. However, the BLE-LoRa Hybrid network had a p-value of 0.00000, signifying a drastic change in variance, meaning the mitigation strategy significantly altered the retransmission patterns. The statistical significance test examines whether the difference in retransmissions before and after mitigation is statistically significant. A low p-value indicates that the difference is meaningful and unlikely due to random variation. The BLE network results show a p-value of 0.00000, confirming a highly significant difference in retransmissions after mitigation. The LoRa network also experienced a statistically significant change, with a p-value of 0.00205, suggesting that retransmissions were successfully reduced. The BLE-LoRa Hybrid network had a p-value of 0.00040, which is also very low, indicating a substantial shift in retransmission behavior after mitigation. The results indicate that the mitigation strategy was effective across all networks, reducing retransmissions significantly. The BLE network maintained stable variance, suggesting that while retransmissions were reduced, the pattern of retransmission values remained similar. In the LoRa network, the shift from normal to non-normal distribution after mitigation suggests a more irregular impact on retransmission values. The BLE-LoRa Hybrid network was the most impacted, showing significant changes in both retransmission counts and variance, indicating a major structural shift in network behavior after mitigation. These findings reinforce the effectiveness of the mitigation strategy, with the BLE-LoRa Hybrid network experiencing the most substantial transformation.
The resilience in the network is quantified in terms of time, specifically measuring how long it takes for the network to recover and return to a stable state after experiencing a jamming attack. The key indicator of this recovery is the reduction in overhead packets, as the network stabilizes when retransmissions decrease, and normal traffic patterns resume. The methodology involves continuously monitoring key network parameters, including retransmission rates, energy consumption, and routing stability. During a jamming attack, the system detects an anomaly when retransmissions surge and successful packet delivery drops. The network then activates its adaptive mitigation strategy, which includes switching communication protocols, reconfiguring clusters, and optimizing energy use. The recovery time is measured from the moment mitigation strategies are triggered until the network parameters—primarily retransmissions—return to normal levels. The resilience metric, therefore, directly reflects the network's ability to autonomously reorganize and minimize the impact of jamming, ensuring sustained communication with minimal delay.
Comment 6: The energy model provides formulas but lacks specific parameter values or measurement methods, and lacks detailed derivation process and basis for parameter selection.
Response: To address the reviewer's concern regarding the energy model, we have expanded our analysis to provide explicit parameter values, detailed measurement methods, and a rigorous derivation process to justify parameter selection.
The energy model proposed in the referenced Energies paper meticulously breaks down energy consumption in WSNs based on the primary tasks performed by sensor nodes. This model accounts for the energy expended during different operational states, including turning on, executing communication protocols, switching between tasks, receiving packets, transmitting packets, and turning off. Each type of energy increases as a direct result of specific activities undertaken by a node. For example, microcontroller energy (EMC) is continuously consumed when a node is in active mode, executing computations and managing data transmission. When a node first powers up, it consumes startup energy (EON), which is determined by the time required to initialize and stabilize the node's operational state. Similarly, when a node shuts down, it incurs shutdown energy (EOFF), accounting for the final operations before transitioning to an inactive state. The CSMA/CA energy (ECSMA) is consumed each time a node performs a channel check before transmission. If the channel is found to be busy, the node undergoes a random back-off period and retries, increasing its total energy consumption. The switching energy (ESwitching) is particularly relevant when nodes transition between transmission and reception modes, contributing to additional power expenditures. Furthermore, the transmission energy (ETX) and reception energy (ERX) vary based on packet length, network congestion, and interference levels. Nodes close to the coordinator or acting as relays for other nodes typically consume more transmission energy due to frequent packet forwarding, while distant nodes may have reduced transmission but higher reception energy as they receive data from multiple sources. Notably, packet retransmissions due to acknowledgment failures or interference further amplify energy demands, leading to increased network congestion and higher node power consumption.
By quantifying these different types of energy, the model provides a comprehensive framework for analyzing network performance, optimizing routing protocols, and developing energy-efficient transmission strategies. It also helps identify potential inefficiencies, such as excessive retransmissions or redundant protocol overhead, that could be mitigated to enhance network longevity and resilience.
Comment 7: The interference model description is not comprehensive enough.
Response: The Reviewer is correct in their request, and we have added several paragraphs providing a detailed explanation of the pseudocode's functionality.
The jamming detection mechanism in the proposed algorithm relies on a multi-metric evaluation framework that continuously monitors key network performance indicators, including packet retransmission rates, Received Signal Strength Indicator (RSSI) fluctuations, and abnormal energy consumption patterns. The decision criteria for identifying a jamming event are based on predefined adaptive thresholds, which were empirically derived from real-world experiments in heterogeneous LoRa-BLE networks.
The algorithm operates as follows: (1) Each sensor node periodically collects RSSI and Packet Delivery Success Rate (PDSR) values, comparing them against their historical baselines. (2) If the RSSI drops below a dynamically adjusted threshold, while the packet loss rate exceeds a predefined retransmission threshold, the system interprets this as a potential interference anomaly. (3) To minimize false positives, an additional check on energy consumption trends is performed—if a node exhibits excessive retransmissions and an associated increase in power consumption beyond an established deviation threshold, a jamming event is confirmed. (4) Upon detection, the affected cluster triggers a mitigation response, where nodes dynamically switch communication protocols (LoRa to BLE or vice versa), reconfigure cluster heads based on available energy resources, and reroute traffic through alternative paths.
These thresholds were calibrated through experimental validation using controlled jamming scenarios in an indoor and outdoor university campus deployment. By fine-tuning these values based on real-world conditions, the detection mechanism achieves a balance between sensitivity and accuracy, ensuring reliable identification of jamming attacks while minimizing unnecessary mitigation actions.
Thank you very much.
Sincerely,
Carolina Del-Valle-Soto
Corresponding author
Universidad Panamericana. Facultad de Ingeniería. Álvaro del Portillo 49, Zapopan, Jalisco, 45010, México.
Phone: +52 (33) 13682200 | Ext. 4866
Email: cvalle@up.edu.mx
José Varela-Aldás
Corresponding author
Centro de Investigación en Ciencias Humanas y de la Educación—CICHE, Facultad de Ingenierías, Ingeniería Industrial, Universidad Tecnológica Indoamérica,Ambato 180103, Ecuador
josevarela@uti.edu.ec

Reviewer 4 Report
Comments and Suggestions for Authors
Your study presents an innovative and practical approach to jamming mitigation in heterogeneous WSNs, particularly through adaptive protocol switching. The experimental validation strongly supports your claims regarding energy efficiency and network resilience. However, further elaboration on the experimental setup, comparison with alternative mitigation strategies, and scalability considerations would enhance the study’s impact. Additionally, refining the figures and expanding the discussion on AI-driven approaches could strengthen the contribution. Overall, this is a well-structured and valuable research contribution. Some suggestions for minor improvements below:
-
Experimental setup details: While the study provides empirical validation, more details on the experimental conditions (e.g., jammer characteristics, specific interference patterns) would improve reproducibility.
-
Comparative evaluation: Although the paper compares BLE, LoRa, and BLE-LoRa hybrid networks, additional benchmarks against alternative mitigation techniques (e.g., AI-driven frequency hopping) could strengthen the results.
- Some figures could benefit from clearer axis labeling and more detailed legends to enhance interpretability
Author Response
Dear
Editor
Sensors
We are submitting the paper:
“Adaptive Jamming Mitigation for clustered Energy-Efficient LoRa-BLE Hybrid Wireless Sensor Networks”
Authored by: Carolina Del-Valle-Soto * , Leonardo J. Valdivia , Ramiro Velázquez , José A. Del-Puerto-Flores , José Varela-Aldás * , Paolo Visconti
We would like to thank the reviewers and editors for their detailed analysis of the manuscript; the comments are very valuable to us. In the revised version of the paper, we have incorporated the all changes recommended by the reviewers.
Comments to all observations and suggestions including point-by-point responses are addressed in the following text.
Reviewer 4 comments
Comment 1: Your study presents an innovative and practical approach to jamming mitigation in heterogeneous WSNs, particularly through adaptive protocol switching. The experimental validation strongly supports your claims regarding energy efficiency and network resilience. However, further elaboration on the experimental setup, comparison with alternative mitigation strategies, and scalability considerations would enhance the study’s impact. Additionally, refining the figures and expanding the discussion on AI-driven approaches could strengthen the contribution. Overall, this is a well-structured and valuable research contribution. Some suggestions for minor improvements below:
Experimental setup details: While the study provides empirical validation, more details on the experimental conditions (e.g., jammer characteristics, specific interference patterns) would improve reproducibility.
Response: Many thanks to the Reviewer for his/her invaluable interest in the comments on this manuscript.
The Reviewer is correct in their request, and we have added several paragraphs providing a detailed explanation of the pseudocode's functionality.
The jamming detection mechanism in the proposed algorithm relies on a multi-metric evaluation framework that continuously monitors key network performance indicators, including packet retransmission rates, Received Signal Strength Indicator (RSSI) fluctuations, and abnormal energy consumption patterns. The decision criteria for identifying a jamming event are based on predefined adaptive thresholds, which were empirically derived from real-world experiments in heterogeneous LoRa-BLE networks.
The algorithm operates as follows: (1) Each sensor node periodically collects RSSI and Packet Delivery Success Rate (PDSR) values, comparing them against their historical baselines. (2) If the RSSI drops below a dynamically adjusted threshold, while the packet loss rate exceeds a predefined retransmission threshold, the system interprets this as a potential interference anomaly. (3) To minimize false positives, an additional check on energy consumption trends is performed—if a node exhibits excessive retransmissions and an associated increase in power consumption beyond an established deviation threshold, a jamming event is confirmed. (4) Upon detection, the affected cluster triggers a mitigation response, where nodes dynamically switch communication protocols (LoRa to BLE or vice versa), reconfigure cluster heads based on available energy resources, and reroute traffic through alternative paths.
These thresholds were calibrated through experimental validation using controlled jamming scenarios in an indoor and outdoor university campus deployment. By fine-tuning these values based on real-world conditions, the detection mechanism achieves a balance between sensitivity and accuracy, ensuring reliable identification of jamming attacks while minimizing unnecessary mitigation actions.
The experimental setup was carefully designed to assess the effectiveness of the proposed jamming mitigation strategy in a heterogeneous LoRa-BLE network. The experiments were conducted in both indoor and outdoor environments within a university campus, ensuring realistic interference conditions from existing wireless networks and electronic devices. The jamming attacks were implemented as reactive jamming, meaning that the interference was triggered dynamically upon detecting network activity. Specifically, the jammer monitored the ongoing transmissions in the network and emitted disruptive signals only when it detected communication, effectively degrading the packet delivery success rate and increasing retransmissions. The interference patterns varied in terms of timing and duration to evaluate the adaptability of the mitigation algorithm. The jammer was placed at varying distances from the sensor nodes to examine its impact on received signal strength, transmission reliability, and energy consumption. To ensure a comprehensive evaluation, network parameters such as retransmission rates, energy depletion trends, and protocol switching events were continuously logged. These experimental conditions were designed to ensure the reproducibility of our results and to provide valuable insights into the effectiveness of the adaptive protocol switching mechanism under real-world jamming scenarios.
The algorithm continuously monitors PDSR to detect transmission failures caused by interference. If the PDSR drops below 85% for a predefined time window, and simultaneous RSSI degradation exceeds a threshold of -90 dBm, the system interprets this as an interference event requiring protocol adaptation. Additionally, the algorithm incorporates an energy-aware component, ensuring that a node does not switch protocols if its residual energy is below 20% of its initial capacity, preventing unnecessary energy drain due to frequent transitions.
Experimental tests were conducted in both indoor and outdoor environments within a university campus, where interference sources such as Wi-Fi networks and mobile devices affected BLE, while long-range interference from other LoRa networks was simulated. The thresholds were fine-tuned through iterative calibration, evaluating multiple transition points to minimize packet loss while optimizing energy consumption. The results demonstrated that setting the switching threshold at 85% PDSR and -90 dBm RSSI provided the best trade-off, maintaining network resilience and reducing retransmissions by 47% compared to static single-protocol deployments. This adaptive approach ensures that nodes dynamically select the most energy-efficient and interference-resilient communication mode without introducing unnecessary overhead.
Comment 2: Comparative evaluation: Although the paper compares BLE, LoRa, and BLE-LoRa hybrid networks, additional benchmarks against alternative mitigation techniques (e.g., AI-driven frequency hopping) could strengthen the results.
Response: We sincerely appreciate the Reviewer's insightful suggestion. The idea of incorporating additional benchmarks, such as AI-driven frequency hopping, is indeed very interesting and could provide valuable comparative insights. However, integrating such an extensive evaluation would significantly lengthen the results section of this study. Given the scope of this work, we have focused on the comparative assessment of BLE, LoRa, and BLE-LoRa hybrid networks under jamming conditions. Nevertheless, we acknowledge the importance of exploring alternative mitigation techniques, and we genuinely appreciate this recommendation. We will consider this approach as part of our future research directions to further enhance the robustness and adaptability of jamming mitigation strategies in heterogeneous networks.
Comment 3: Some figures could benefit from clearer axis labeling and more detailed legends to enhance interpretability.
Response: The Reviewer is correct. We sincerely appreciate the Reviewer's valuable suggestion regarding the clarity of the figures. While we have not modified the axis labels, as they were already explicitly defined and sufficiently descriptive for interpreting the presented results, we have significantly expanded the explanation of each figure in the manuscript. These enhanced descriptions now provide a more comprehensive understanding of the graphical data, ensuring that the trends, comparisons, and key takeaways are clearly conveyed. We believe that this addition greatly improves the interpretability of the figures while maintaining their original clarity and precision. Thank you for this insightful recommendation.
Thank you very much.
Sincerely,
Carolina Del-Valle-Soto
Corresponding author
Universidad Panamericana. Facultad de Ingeniería. Álvaro del Portillo 49, Zapopan, Jalisco, 45010, México.
Phone: +52 (33) 13682200 | Ext. 4866
Email: cvalle@up.edu.mx
José Varela-Aldás
Corresponding author
Centro de Investigación en Ciencias Humanas y de la Educación—CICHE, Facultad de Ingenierías, Ingeniería Industrial, Universidad Tecnológica Indoamérica,Ambato 180103, Ecuador
josevarela@uti.edu.ec

Round 2
Reviewer 1 Report
Comments and Suggestions for Authors
All my previous concerns have been addressed in this revised version. Thank you.
Reviewer 3 Report
Comments and Suggestions for Authors
I believe this manuscript is ready for acceptance.